# Agro-Industrial Wastewaters for Algal Biomass Production, Bio-Based Products, and Biofuels in a Circular Bioeconomy

**Júlio Cesar de Carvalho ***, **Denisse Tatiana Molina-Aulestia, Walter José Martinez-Burgos, Susan Grace Karp, Maria Clara Manzoki, Adriane Bianchi Pedroni Medeiros , Cristine Rodrigues, Thamarys Scapini , Luciana Porto de Souza Vandenberghe, Sabrina Vieira , Adenise Lorenci Woiciechowski, Vanete Thomaz Soccol  and Carlos Ricardo Soccol ***

Bioprocess Engineering and Biotechnology Department, Federal University of Paraná—Polytechnic Center, Curitiba 81531-980, Brazil

*   Correspondence: jccarvalho@ufpr.br (J.C.C.); soccol@ufpr.br (C.R.S.)

**Abstract:** Recycling bioresources is the only way to sustainably meet a growing world population's food and energy needs. One of the ways to do so is by using agro-industry wastewater to cultivate microalgae. While the industrial production of microalgae requires large volumes of water, existing agro-industry processes generate large volumes of wastewater with eutrophicating nutrients and organic carbon that must be removed before recycling the water back into the environment. Coupling these two processes can benefit the flourishing microalgal industry, which requires water, and the agro-industry, which could gain extra revenue by converting a waste stream into a bioproduct. Microalgal biomass can be used to produce energy, nutritional biomass, and specialty products. However, there are challenges to establishing stable and circular processes, from microalgae selection and adaptation to pretreating and reclaiming energy from residues. This review discusses the potential of agro-industry residues for microalgal production, with a particular interest in the composition and the use of important primary (raw) and secondary (digestate) effluents generated in large volumes: sugarcane vinasse, palm oil mill effluent, cassava processing waster, abattoir wastewater, dairy processing wastewater, and aquaculture wastewater. It also overviews recent examples of microalgae production in residues and aspects of process integration and possible products, avoiding xenobiotics and heavy metal recycling. As virtually all agro-industries have boilers emitting $CO_2$ that microalgae can use, and many industries could benefit from anaerobic digestion to reclaim energy from the effluents before microalgal cultivation, the use of gaseous effluents is also discussed in the text.

**Keywords:** POME; vinasse; cassava; chlorella; anaerobic digestion; waste treatment

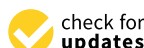



## 1. Introduction

The beginnings of modern microalgae biotechnology were surrounded by significant excitement because of the enormous potential extrapolated from the high productivity achieved in laboratories. However, while the search for algal biofuels since the 1980s has evolved a lot, it has not been able to consistently provide products to the market. Conversely, nutritional and nutraceutical biomass products have flourished, as shown by the FDA's GRAS certifications (generally recognized as safe certifications by the U.S. Food and Drug Administration) and the growing offer of consumer products. The market for microalgae products today has wildly diverse estimates, but on the same order of magnitude of USD 1 to 5 billion by 2027 with a CAGR (compound annual growth rate) of 4–7% [1,2]. Even if the market remains in the lower figures, the projected growth is enormous and is driving research and development in the area.

The unsuccess of biofuels relative to specialty products does not mean that fuels have not evolved. However, their sales value is not yet capable of covering production costs.

The use of agro-industrial residues, produced in huge volumes in agro-industrial and agro-energy processing, could be a way of enabling biofuel production. The idea is to couple a need with an opportunity: the need to carry out biological treatments of agro-industrial liquid effluents and returning clean water to the environment and the opportunity of reclaiming nutrients and water by cultivating microalgae culture in these residues, in raw or minimally treated form [3–5].

Liquid agro-industrial effluents are processing and washing waters produced during the industrialization of energy or food crops or livestock. Today, most waste undergoes simple biological treatment with the eventual production of biogas and the use of the final liquid effluent for fertigation [6–8]. However, this process does not entirely take advantage of the large amount of nutrients available in the waste. The final steps of biological treatment, namely stabilization ponds, typically already have native microalgae growing [9,10]. Agro-industry residues are produced in large amounts and have low toxicity compared to other industrial effluents but also have a high eutrophication potential because of the availability of nutrients, especially nitrogen and phosphorus.

Effluents produced in small amounts can also be used for microalgal cultivation and the logic is the same—taking advantage of the leftover, diluted nutrients. However, small processing plants have a proportionally higher CAPEX (capital expenditures) than large-scale plants. That is why most interest goes to a few effluents produced in staggering amounts, such as vinasses. The large volume of residues can be inferred from agro-industrial production [7]: in 2020, crop production worldwide was estimated at 9.3 billion tons, including food, energy, and animal feed crops [11] (Figure 1). Some of these crops, such as potatoes and tomatoes, are only minimally or partially processed and others do not generate large volumes of liquid waste, such as cereals. However, much of cassava, palm, and sugarcane are extensively processed into starch, palm oil, and ethanol, generating large volumes of waste.

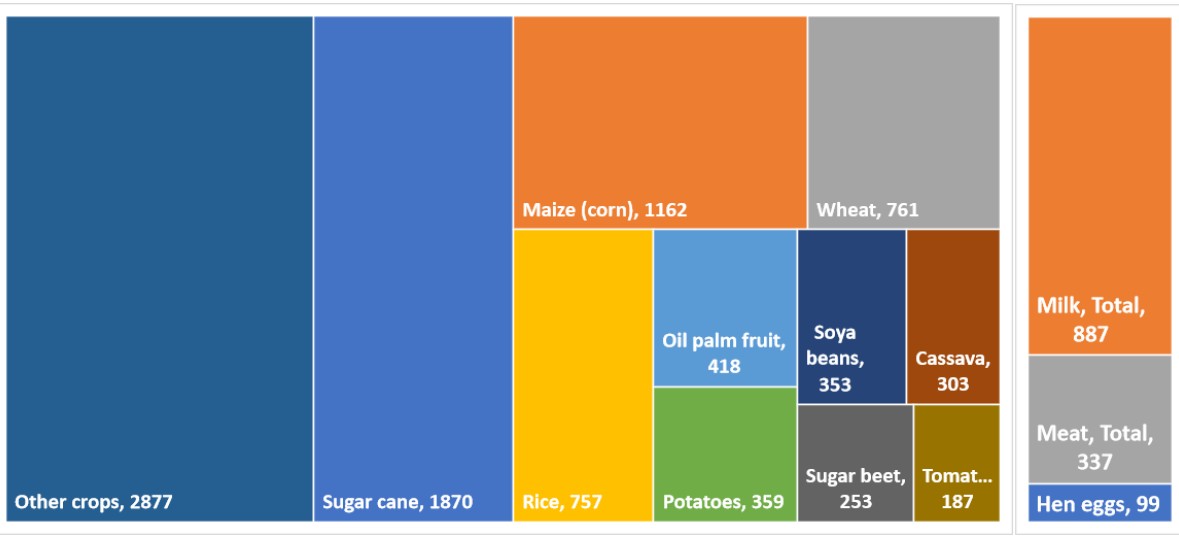

**Figure 1.** Food and energy crops and animal products in the world in 2020, in million tons. Data compiled from the FAO database [11]. Egg production was converted assuming 60 g/egg.

Liquid agro-industry wastes are usually too diluted for burning or composting but are also too concentrated to be directly discarded. Microalgae can be cultivated in such residues [12–14] after residue pretreatment, microalgae adaptation, or both. Three main effluent streams could be available from industries: two liquid and one gaseous fraction. The two liquid fractions are (1) the primary effluents, derived directly from the process (wash waters and residue streams, such as vinasses from ethanol production), with high COD (chemical oxygen demand), and (2) the secondary effluents, the product of the previous treatment of the primary effluents. This treatment can be aerobic, but because

of the high CODs, it is more common to use anaerobic digestion. Therefore, secondary effluents are also termed *digestates*. Secondary effluents usually have a lower COD since part of it was removed during the primary treatment. Both residues have variable concentrations of several macro- and micronutrients. The gaseous effluent fraction—commonly called flue gases—is the product of the combustion of fuels that heat boilers and generate steam. Flue gases are composed of air, $CO_2$, and minor amounts of $NO_x$, $SO_x$, and CO. These gases cannot sustain microalgal growth alone but can be a source of $CO_2$ [15]. A final source of $CO_2$ for microalgae cultivation in circular processes could be the biogas produced in anaerobic digestion, which is not a residue but can be upgraded to biomethane by removing the $CO_2$.

Microalgae can produce a multitude of products of industrial interest, from biomass and nutraceutical fractions to biofuels; selecting an appropriate product can add value to waste and increase the industry income and the circularity of the process. By providing nutrients at a meager cost, using residues can reduce expenses in algal production. This is advantageous for already established products, such as PC (phycocyanin, a blue-colored protein from cyanobacteria) and AX (astaxanthin, a carotenoid), and enables the production of new products, such as biofuels and PHAs (polyhydroxyalkanoates, biopolymers produced by some cyanobacteria) [16–18].

Some bottlenecks to the massive production of microalgal biomass, such as the low cell concentration in cultures and the consequent need for large cultivation areas and large volumes of culture media and nutrients, can be solved using agro-industry wastewaters. However, these residues bring other challenges, such as the high COD and its sometimes-intense color and turbidity, which can reduce light availability. As a result, microalgal cultures—rarely axenic—can be dominated by bacteria and other microorganisms that compete for nutrients or even predate microalgae [19]. This means that pre-treatments may be necessary to ensure high productivity. Microalgae selection and adaptation are also required.

Nevertheless, the advantages of including a microalgal production step in biorefineries are many and can overcome various difficulties, as can be perceived from the intensive recent research in the area. Recycling bioresources, reducing COD through microalgal cultivation, reusing water, producing new products, and creating value through a circular approach are tendencies in bioprocesses. The following sections discuss the most relevant aspects of the use of agro-industrial residues in the production of microalgae products, starting with an overview of larger-volume residues—those that are produced in a more concentrated and favorable way for the development of large-scale processes. The text proceeds to describe successful cases and approaches in producing microalgae in these residues. Then, it discusses the important issue of using gaseous effluents as a carbon source and as an opportunity to mitigate emissions from the agro-industry. Finally, it presents possible algal products that can be targeted for integrated processes and discusses new process development and future perspectives.

## 2. Agro-Industrial Wastewaters Overview

Liquid effluents can be efficiently used as culture media for microalgae because of their nutrient availability, comparable to that of classical media, such as Zarrouk, BBM (Bold's basal medium), or BG-11 (blue-green medium). One of the main difficulties in the cultivation of microalgae on a large scale is the high consumption of water and nutrients for medium, which leads to high costs [20], so the use of effluent could be a strategy for the dissemination of microalgae cultivation and sustainability in production processes [21,22].

Agro-industrial effluents are wastewaters generated by processing raw materials from agriculture, forestry, or fishing. They can be generated as a by-product of the process or derived from the cleaning procedures of lines and equipment. Among the effluents that are generated as by-products are cassava processing wastewater (CPW), palm oil mill effluent (POME), sugarcane vinasse (SV), abattoir wastewater (AW), aquaculture wastewater (AqW), and dairy wastewater (DW).

Sugarcane vinasse (SV) is wastewater generated through the separation of alcohol from the fermentation broth [7]. According to [23], sugarcane vinasse is generated in a proportion of 12–15 $m^3/m^3$ ethanol. Considering that, in 2021, 124.15 million $m^3$ of bioethanol were produced worldwide [24], it can be inferred that around 1.86 billion $m^3$ of vinasse were generated. In the case of POME and CPW, 3.45 tons POME/ton of oil and 0.6 $m^3$ CPW/ton of cassava were generated, respectively; thus, it can be inferred that 285.45 million tons of POME and 75.6 million $m^3$ CPW were generated in the year 2021 [25,26].

Global meat production in 2020 reached almost 337.18 million tonnes for cattle, poultry, pigs, and sheep, according to the meat statistics from the United Nations Food and Agriculture Organization, 2021. About 27% of the freshwater used by humans is consumed by animal production. Meat production requires between 6 and 20 times more water than cereals and vegetables or fruits and generates from 1.5 to 18 $m^3$ of effluents/ton of meat produced [27]. Meat processing plants produce large quantities of abattoir wastewater (AW) from slaughtering and processing animals and cleaning the facilities. The effluent composition varies by type of animal and process specificities [28]. Organic matter, suspended particles, oils and greases, blood, manure, urine, and meat tissues make up the effluent from slaughterhouses. Blood is one of the main dissolved contaminants, and its separation efficiency in processes is essential for ensuring lower effluent loads [29]. This effluent is characterized by high BOD, COD, high oil concentrations, total solids, total suspended solids, total nitrogen, and chlorides [30]. Nitrogen compounds are one of the main nutrients in wastewater; nitrates accelerate plant growth in water bodies and can be easily reduced to nitrites, threatening human health and the environment when treated incorrectly [31].

The growing demand for dairy products has resulted in the development of many dairy industries, generating wastewater with high polluting potential [32]. It is estimated that world milk production will grow by 1.6% per year over the next decade (reaching about 997 Mt by 2029), and the world per capita consumption of fresh dairy products is projected to increase by 1.0% per year by 2030. Wastewater in the dairy industry (DW) is composed of carbohydrates, proteins, grease or oil, and high phosphate and nitrate concentrations [33]—so much so that in a *Tetradesmus obliquus* culture, the substitution of 20% (*v/v*) of Bold's basal medium (BBM) by acid whey permeate was considered the best compensation for biomass growth, enzyme production, and nutrient utilization [34].

Aquaculture will account for 52% of all fish production by 2030, reaching 105 Mt, according to a report by the Organization for Economic Co-operation and Development (OECD)/Food and Agriculture Organization (FAO) [35]. Aquaculture wastewater (AqW) comprises dissolved and particulate organic matter and dissolved solids, such as phosphorus and nitrogen compounds. Food and feces are present in aquaculture effluents and cause environmental deterioration of both receiving water bodies and sediments. They cause the increased metabolic activity of aerobic bacteria, and the scarcity of oxygen by bacterial action leads to the destruction of the most vulnerable aerobic life forms in rivers and lakes [36].

Wastewater is a major concern in the world due to the negative aspects and impacts it can generate on the environment. The effects of agro-industrial effluents on the environment depend mainly on their physical-chemical composition. In general, agro-industrial effluents have high organic loads (measured as COD), reducing the amount of dissolved oxygen in water bodies if incorrectly disposed of, thus affecting aquatic ecosystems. The high organic loads of agro-industrial effluents are explained by high concentrations of carbohydrates, proteins, and even fats [7]. These effluents are also a significant source of macronutrients, such as nitrogen, phosphorus, and potassium (Table 1).

According to [37,38], excess nitrogen and phosphorus in water bodies induce eutrophication processes, that is, the accelerated growth of plants or microalgae that affect the balance of the aquatic ecosystem. Agro-industrial effluents are also a significant source of micronutrients, such as chromium, cobalt, copper, iron, zinc, manganese, molybdenum,

nickel, and selenium. These compounds are present in small amounts in wastewater. However, they are essential in the catabolic reactions developed by microorganisms. SV, for example, is a source of different micronutrients, including Co, Fe, Mo, Se, and Zn. POME and CPW are also sources of micronutrients, such as Fe, Zn, and Mn.

These effluents are also a source of amino acids, such as proline, glycine, alanine, cysteine, valine, methionine, isoleucine, leucine, tyrosine, phenylalanine, histidine, lysine, and arginine. [39–41]. The presence of amino acids in effluents can improve microbial growth or the production efficiency of some byproducts, as they can alter metabolic pathways [42].

Agro-industrial wastewaters also contain significant amounts of toxic compounds, such as heavy metals, which can bioaccumulate in the food chain. For example, SV contains heavy metals in significant concentrations, such as Cd, Cu, Cr, Ni, and Pb, in concentrations of 200 ppb, 200 ppb, 100 ppb, 2000 ppb, and 5000 ppb, respectively. [43]. Other metals, such as As, Hg, and Zn, have also been found in this effluent [44]. Metals, such as Cd, Cr, Hg, and Cu, have also been found in POME and CPW effluents. [45,46]. In the case of CPW, it also contains significant amounts of cyanide, which is highly toxic [18]. While microalgae can transform low-cost raw nutrients from wastewater into valuable biomass components [47], the excess of certain metals can be harmful or lead to bioaccumulation, as discussed in Section 7 of this text.

**Table 1.** Physical-chemical composition of agro-industrial effluents that are produced in large volumes.

| | Parameter | SV, Sugarcane Vinasse | POME, Palm Oil Mixed Effluent | CPW, Cassava Processing Wastewater | AW, Abattoir Wastewater | DW, Dairy Wastewater | AqW, Aquaculture Wastewater |
|---|---|---|---|---|---|---|---|
| Main effluent characteristics | COD (g$O_2$ L$^{-1}$) | 27.7–299.5 | 22.65–85.71 | 1.4–141.3 | 190–12800 | 91.6–4000 | 6.1–165 |
| | BOD$_5$ (g$O_2$ L$^{-1}$) | 5.04–47.63 | 34.71–52.67 | 1.96–44.6 | 53–2250 | 90–1814 | 8.5 |
| | pH | 4.04–6.5 | 4.3–5.2 | 3.9–7.2 | 6.4–7.7 | 4.33–7.35 | 7–8.2 |
| | Total solids (g L$^{-1}$) | 7.5–36.6 | 47–64.6 | 5.4–92.9 | - | 10.8–9290 | - |
| | Fixed solids (g L$^{-1}$) | 0.27–14.16 | - | | - | 2594 | - |
| | Volatile solids (g L$^{-1}$) | 4.7–26.3 | 24.84–30.92 | 3.5 | - | - | - |
| Nutrients | Ammonia (mg L$^{-1}$) | 18–118.1 | 77–101 | 61.53 | 6.5–532.3 | 53–115 | ≤0.1–6.25 |
| | TKN (mg L$^{-1}$) | 122–540 | 180–1400 | 180 | | 22–2209 | |
| | Nitrate (mg L$^{-1}$) | 0,1–45.3 | 109–136 | 4 | 11.97–140 | 18.05 | 0.35–152.8 |
| | Nitrite (mg L$^{-1}$) | 0.1–0.4 | - | - | 13.7–54 | 0.398 | 0.3–24.7 |
| | Sulfate(mg L$^{-1}$) | 669–3070 | - | 63.77 | 12.63–500 | 17 | 420.6 |
| | Phosphorus (mg L$^{-1}$) | 44–232 | 109–136 | 0.4 -3000 | 7–108 | 11.6–3055 | 16.1 |
| | Magnesium (mg L$^{-1}$) | 343–616 | 279–296 | 17.68–43.08 | 18.64 | 18 | 39.6–49.75 |
| | Calcium (mg L$^{-1}$) | 344–609 | 282–290 | 18.41–86.49 | 4.9–67 | 129 | 168.9–118 |
| | Potassium (mg L$^{-1}$) | 1542–3652 | 1696–2043 | 15.92–435.75 | 8.1–90 | - | 195.1 |
| | Sodium (mg L$^{-1}$) | 27–57 | 94–113 | 16.42–17.68 | 621 | - | 246.4 |
| | Fluoride(mg L$^{-1}$) | 0.14–0.44 | - | | - | - | |
| | Chloride (mg L$^{-1}$) | 209–3548 | 94–113 | 52 | 352 | 1204 | 147.3 |
| Metals | Aluminum (mg L$^{-1}$) | 1.13–11.9 | - | 1.12 | - | - | - |
| | Barium (mg L$^{-1}$) | 0.23–0.56 | - | - | - | - | - |
| | Cadmium (mg L$^{-1}$) | 0.08–0.027 | 0.01–0.02 | - | 0.009 | - | - |
| | Arsenic (mg L$^{-1}$) | 0.098–0,14 | - | - | - | - | - |
| | Chrome (mg L$^{-1}$) | 0.028–0.084 | 0.05–0.43 | - | - | - | - |
| | Cobalt (mg L$^{-1}$) | 0.011–0.035 | 0.04–0.06 | - | - | - | - |
| | Copper (mg L$^{-1}$) | 0.19–1.16 | 0.8–1.6 | 3.26 | - | - | - |
| | Iron (mg L$^{-1}$) | 5.8–18.6 | 65–164 | 72,65 | 0.9–21 | - | - |
| | Lead (mg L$^{-1}$) | 0.01–0.59 | | 0.72 | 0.5–4 | - | - |
| | Manganese (mg L$^{-1}$) | 1.04–4.62 | 2.1–4.4 | 3.3 | - | - | - |
| | Mercury (mg L$^{-1}$) | 0.01 | - | - | - | - | - |
| | Molybdenum (mg L$^{-1}$) | 0.016–0.066 | - | - | - | - | - |
| | Nickel (mg L$^{-1}$) | 0.038–0.12 | 0.1–3.6 | - | - | - | - |
| | Selenium (mg L$^{-1}$) | 0.02–0.076 | | - | - | - | - |
| | Zinc (mg L$^{-1}$) | 0.2–1.19 | 1.2–2.72 | 25,11 | 0.178 | - | - |

**Table 1.** *Cont.*

| | Parameter | SV, Sugarcane Vinasse | POME, Palm Oil Mixed Effluent | CPW, Cassava Processing Wastewater | AW, Abattoir Wastewater | DW, Dairy Wastewater | AqW, Aquaculture Wastewater |
|---|---|---|---|---|---|---|---|
| Compounds | Lipids (g L$^{-1}$) | - | 8.81–37.88 | - | - | 34.217 | - |
| | Glycerol (mg L$^{-1}$) | 3333 | - | - | - | - | - |
| | Reducing sugar (mg L$^{-1}$) | - | 228–236 | 1300 | - | - | - |
| | Cyanide (mg L$^{-1}$) | - | - | 46,75 | - | - | - |
| References | | [40,44] | [48–50] | [51–53] | [28–31,54–57] | [58–62] | [63–67] |

BOD$_5$, biochemical oxygen demand in 5 days of incubation; TKN, total nitrogen measured via the Kjeldahl method.

## 3. Direct Microalgae Cultivation in Residues—Description and Examples

### 3.1. Vinasse

The utilization of vinasse for growing microalgae is limited due to the high color and turbidity of this agro-industrial effluent, which can reduce light penetration in the algal culture, reducing photosynthesis [68]. A common practice consists of diluting the vinasse to enable microalgae cultivation [69–71], although the use of diluted vinasse as a culture medium is not a desirable feature at the industrial scale [72]. These issues can also be solved by clarifying the vinasse using chemical clarifying agents, such as Al$_2$(SO$_4$), FeCl$_3$, and Ca(OH)$_2$ [73], or natural alternatives, such as *Moringa oleifera* seeds with activated charcoal [68]. Other alternatives to clarify vinasse are the mechanical filtration, centrifugation, decantation, and decomposition of the organic matter using electrochemical, oxidative, and biological means [74]. These methods, however, are rarely applied on a large scale due to the related costs [68]. Another well-established treatment for vinasse before microalgae cultivation is anaerobic digestion (AD). It is usually carried out in up-flow anaerobic sludge blanket (UASB) reactors since they can be cost-effective due to simple operation and high efficiency in removing the organic load. Moreover, AD allows biogas collection, which can be used both for its energetic and CO$_2$ content [75]. Usually, vinasse goes through a pretreatment before entering the UASB, when the acidic pH is adjusted to ensure the stability of the anaerobic digestion process [76]. Additionally, the removal of suspended solids from the vinasse before AD may be necessary to avoid system acidification due to the excess of volatile fatty acids (VFAs) [75]. In the study by [77], this strategy was used to cultivate *Chlorella vulgaris,* reaching a productivity of 70 mg L$^{-1}$ day $^{-1}$. In this study, the combination of anaerobic digestion and *C. vulgaris* cultivation showed a significant reduction of vinasse organics, N, and P contents with a COD and turbidity removal greater than 80%.

Table 2 presents other microalgae cultivated in raw or highly concentrated vinasse media. Some studies demonstrate higher productions of chlorophyll a and b as well as proteins and carbohydrates associated with the presence of determined percentages of vinasse in the media [78,79]. Inoculating a microalgae culture typically cultivated in synthetic media into the agro-industrial residue can lead to a culture crash [3]. This is due to the high concentrations of certain compounds in effluents, which can damage the microalgae osmoregulation and metabolic processes [80]. It is necessary, thus, to slowly adapt the cultures through sequential inoculations into media with increasing effluent concentrations, as was the case, for example, of the study of Sydney et al. [81], in which microalgae of the genus *Botryococcus, Synechococcus, Chlorella, Neochloris, Scenedesmus*, and *Arthrospira* were inoculated initially in 10% vinasse media and afterward in 20% and 30% concentrations. High inoculum concentrations can also help with culture competition, and previous adaptation to external conditions can be made before inoculation into large volumes of effluent [82].

### 3.2. POME, Palm Oil Mixed Effluent

POME usually contains high amounts of suspended solids that can adversely affect microalgae cultivation because of low light penetration. Particulate matter can also reduce nutrient availability to the cells due to the adsorption or absorption of nutrients [83,84].

Similarly to other wastewaters, POME may contain inhibitors, such as excessive amounts of residual oil, which can lead to microalgae sedimentation and, consequently, a reduction in treatment efficiency due to biomass losses [85]. For those reasons, pretreatments such as anaerobic digestion of POME may be critical before microalgae cultivation [83], with AD eventually followed by centrifugation for solids [86]. To improve light access during cultivation, processes such as coagulation or absorption may be considered. Adsorption can use activated carbon extracted produced in situ from palm kernel shells [87,88]. Acid–heat treatment is another option, being especially useful for reducing the dark color observed in the effluent due to lignin removal and also increasing the concentration of sugars that can contribute to mixotrophic microalgae growth [87]. Overall, filtration, autoclaving, or exposure to UV light [89] can be performed to treat the wastewater regarding microbiological levels before microalgae cultivation.

Another option to treat wastewater is to utilize immobilized microalgae. It is the case of Immobilized *Nannochloropsis oculata* in down-flow hanging sponge (DHS) reactors for treating POME [90]. The efficiencies of removal of COD, ammoniacal nitrogen ($NH_3$-N), and phosphate were of 73%, 80%, and 83%, respectively, with high color removal as well. Only minerals ($CaCl_2.2HO$, $MgSO_4.7H_2O$, $Fe_2SO_4.5H_2O$), trace elements ($CuSO_4$, $NiCl_2.6H_2O$, $CoCl_2.6H_2O$), and phosphate buffer ($KH_2PO_4$, $K_2HPO_4$) were supplemented to the microalgae to guarantee the pollutants removal [90]. In another study, immobilized *Chlorococcum oleofaciens* in alginate beads (as in the previously mentioned case) were able to remove total phosphorus, nitrogen, ammonia nitrogen, and soluble chemical oxygen by 90.43%, 93.51%, 91.26%, and 50.72%. However, during the process, the beads suffered degradation; thus, COD values increased, showing that bead stability is an important factor in the treatment [91]. Immobilized microalgae can also be used to treat other types of agro-industrial effluents. It is the case of *Synechococcus pevalekii*, which was used to treat rice vinasse, a decrease in nitrate content and total solid removal of 91 and 29% was observed after 10 days, with significant $Mg^{2+}$, $K^+$, $Cl^-$, and $PO_4^{3-}$ removals. Subsequently, the treated vinasse was used as culture media (69% *v/v*) for cultivating *Dunaliella salina* in mixotrophic cultivation, with a 175% increase in cell number when compared to 100% Conway medium [92].

*3.3. CPW, Cassava Processing Wastewater*

The use of cassava processing waste (CPW) as a culture medium for microalgae production has increased due to the volume generated and its nutritional composition, allowing the production of various microalgae and metabolites [93]. It presents high levels of carbohydrates, proteins, and fats, with high organic loads—the COD is, on average, 55 g $O_2$ $L^{-1}$ and the BOD (biochemical oxygen demand) is 10 g $O_2$ $L^{-1}$ [7].

Due to the high concentration of some components, such as phosphate and cyanide (Table 1), it is not always possible to CPW directly. Typical CPW treatment consists of sieving, primary sedimentation (separation of sand and other particles), and an anaerobic lagoon and facultative lagoon [18]. Currently, processes such as filtration, flocculation, and sterilization are also being evaluated as pretreatment for this effluent. However, the most promising current research is using the anaerobic digestion of CPW [17,94]. After pretreatment, the effluent CPW (or its digestate) still contains high amounts of nitrogen, phosphate, and soluble sugars, thus allowing the mixotrophic or heterotrophic microalgae production for bioproducts, such as biomass, biodiesel, or pigments [95]. Table 2 describes several products produced using CPW as a substrate.

Similarly to CPW, the cultivation of microalgae in CPW requires an adaptation stage of the culture. The most common conditions are (I) the dilution of the effluent (challenging the culture with growing residue concentrations) and (II) the isolation and selection of resistant strains. For instance, four possible strains of *Chlorella* sp., *Scenedesmus* sp., *Monoraphidium* sp., and *Golenkinia* sp. were isolated from a secondary effluent of a cassava processing industry and evaluated using non-sterile CPW, sterile CPW, and pre-treated CPW by anaerobic digestion at different concentrations. The crude CPW is not viable;

diluted CPW could be favorable. In the case of sterilized CPW, the appropriate dilution was 30%, while in the case of pure digested CPW, the growth of *Monoraphidium* sp., *Golenkinia* sp., and *Scenedesmus* sp. was identified. For *Chlorella* sp., the adequate dilution was 40%. The cell values obtained were $2 \times$, $4 \times$, and $7 \times 10^5$ cells mL$^{-1}$ for *Monoraphidium* sp., *Golenkinia* sp., and *Scenedesmus* sp., respectively, and $1 \times 10^5$ for *Chlorella* sp., [18].

The dilution adaptation process can be carried out using synthetic culture media or water. In a process development for *Haematococcus* production, BBM was used until the culture reached the exponential phase. Then, for mixotrophic and heterotrophic growth, CPW was added to the BBM medium [96].

### 3.4. Abattoir Wastewaters

Most studies of microalgae cultivation in slaughterhouse effluents were performed after anaerobic digestion. Acid treatment for precipitation is another pretreatment option: by reducing the pH value of the effluent from 6–7 to 4, about 80% of the total COD was removed as sludge [97]. Afterward, *Chlorella vulgaris* was cultivated in the effluent and 83% of the remaining COD was removed, reaching a biomass concentration of 1.2 g L$^{-1}$. Microalgae can be especially useful for removing the high ammonium and phosphate contents in slaughterhouse effluents [98], and parameters such as the pH of the wastewater can be modified to suit the microalgae of interest, as was the case in the study by Katırcıoğlu et al. [98], in which effluent pH values from 9 to 11 were tested. The pH of 10.5 was considered the best for optimizing nutrient removal and obtaining total lipid and fatty acid methyl esters in *Chlorella vulgaris* comparable to cultivation in BG-11. Moreover, this alkaline pH inhibited the presence of rotifers. At the end of the twelfth day, ammonium and phosphate removals were 99% and 96.43%, respectively [98]. In the study by [99], the filamentous microalgae *Stigeoclonium* FAUBA-10 achieved biomass productivity of 0.45 g L$^{-1}$ d$^{-1}$ in semicontinuous cultivation in slaughterhouse effluent, with a nutrient removal above 92% in 5 days. This microalga also has the operational advantage of easy harvesting by filtration through mesh screens (<341 μm). Aside from pH, $CO_2$ injection may play an important role in nutrient removal from ADAE (anaerobically digested abattoir effluent), since it participates in the pH equilibrium in the medium and avoids ammonia stripping caused by the deionization of ammonium ($NH_4^+$) to ammonia ($NH_3$), which, at high concentrations, is toxic to microalgae cells [100,101].

Microalgae typically coexist with certain bacteria species through symbiotic interactions, and this relationship can be significant in wastewaters [102]. Together, both groups can interact and perform better towards environmental stresses. Microbial diversity depends on the wastewater source and environmental conditions [80]. When treating dairy farm and poultry slaughterhouse wastewaters, the microalgae *Tetradesmus obliquus* and the bacterium *Variovorax paradoxus* were co-cultivated. They attained higher microalgal growth with efficient nutrient removal and the highest reduction in COD (83–87%) compared to mono or sterile cultivations. It was evident that not only this microalgal–bacterial interaction happened but further interactions with other native bacterial communities in the tested wastewaters also increased bioremediation [80]. Additionally, the microalgae *Coelastrella* sp., *Chlamydomonas* sp., and *Scenedesmus* sp. were cultivated in POME and showed better phycoremediation in the unsterilized residue due to microalgae–bacteria consortia interaction, with the highest values of COD and $NH_4^+$ removal when compared to sterilized POME. Evidence of shifting in the bacteria community was seen, with Actinobacteria, Bacteroidetes, Planctomycetes, and Proteobacteria being the major phyla found [86].

### 3.5. Dairy Processing Wastewater

The dairy industry is one of the most polluting due to the variety of products generated, i.e., milk, yogurt, cheese, cream, butter, ice cream, and others. For this reason, the composition of the wastewater presents high concentrations of nutrients, fats, chlorides, sulfates, and lactose [103]. For instance, the milk effluent analysis showed COD values of

85–95,000 mg L$^{-1}$, BOD of 40–48,000 mg L$^{-1}$, total nitrogen of 14–830 mg L$^{-1}$, and a total phosphorus of 9–280 mg L$^{-1}$ [104].

Nevertheless, these nutrients are not readily fermentable, so they need processes that facilitate the presence of simple sugars. Some pretreatments described are anaerobic fermentation, filtration, sedimentation, sterilization, ultrasound, and pH variations [105,106]. The effluent from a dairy industry used for *Chlorella* production was pretreated by the addition of NaOH (5M) up to pH 7, heating ($\approx$80 °C), and centrifugation [107]. Other pretreatments include the use of homogenization tanks, which act as grease traps in addition to allowing solids to settle [108].

In addition to pretreatment, inoculum adaptation is necessary before starting microalgae production using effluents. One technique currently used is the isolation of extremophilic microalgae. For instance, potential microalgae were isolated from dairy effluents. During this stage, effluent samples were cultivated in artificial media such as BG-11 or BBM [109,110]. Because many of the processes are performed with commercial strains, an adaptation step is necessary before cultivation. In experiments conducted with *Chlorella sorokiniana* UTEX 1230 (CCAP, Oban, UK), the microalgae were gradually adapted to medium-strength DW digestate for eight weeks. For another six weeks, the proportion of wastewater was increased from 10% to 90% in order to finally use the undiluted effluent [111].

### 3.6. Aquaculture Wastewater

Effluents generated from the aquaculture industry are of great interest due to the high degree of contamination from the excessive use of artificial nutrients, chemicals, and antibiotics. The use of microalgae in this effluent has several benefits (I) treatment of the effluent from the massive production of algae and (II) production of biomass or bioproducts [112] that are of particular interest in the aquaculture industry.

The quantity and quality of this effluent vary according to the species cultivated, the scale of production, the type and location of the system, and the technology used for treatment. In general terms, AW is composed of suspended solids (i.e., fish excreta and feed waste) and inorganic nutrients, such as ammonia ($NH_3$), nitrate ($NO_3^-$), nitrite ($NO_2^-$), and phosphorus (as $PO_4^{3-}$), whose concentrations depend on the type and cycles of production [21]. For example, the average chemical oxygen demand (COD) is approximately 300 mg L$^{-1}$.

The systems most used as pretreatments for this effluent are sedimentation, precipitation to remove phosphorus, and biological denitrification [113]. Due to their characteristics and natural composition, diatoms are used to cultivate fish farm effluents, efficiently eliminating excess nutrients in 3–5 days (open pond cultures) [112].

**Table 2.** Microalgae growth in agro-industrial wastewater.

| Agro-Industrial Effluent | Microalgae | Medium Additives | Productivity (g L$^{-1}$ day$^{-1}$) | Biomass (g L$^{-1}$) | COD Reduction (%) | BOD Reduction (%) | Bio-Products | Reference |
|---|---|---|---|---|---|---|---|---|
| Vinasse | *Chlorella vulgaris* | 10% saline medium is composed by (g L$^{-1}$):(0.25) Mg (NO$_3$)$_2$·6H$_2$O; (0.025) CaCl$_2$·2H$_2$O; (0.0075) MgSO$_4$·7H$_2$O; (0.015) KH$_2$PO$_4$ (0.0025) NaCl and 1 mL L$^{-1}$ of trace elements. 540 mg L$^{-1}$ oxytetracycline 450 mg L$^{-1}$ of ampicillin | 2.1 | 10.5 | 49.1 | 70.0 | Biomass | [114] |
| Vinasse | *Scenedesmus bajacalifornicus* | - | 0.325 | 3.9 | 51.9 | 60.3 | Biomass | [115] |
| Vinasse | *Desmodesmus subspicatus* | - | 1.450 | 2.9 | 66 (TOC*) | - | Biomass | [116] |
| Vinasse | *Arthrospira maxima* | 70% water | 0.150 | 2.25 | 81 | 89.2 | Peptide fractions (57% of proteins in biomass) | [117] |
| Vinasse | *Desmodesmus* sp. | - | 2.426 | 4.0 | 36.2 | - | Biomass | [118] |
| Vinasse | *Chlorella vulgaris* | - | 0.073 | 2.7 | 15.6 | - | Biomass(10% of lipids in biomass) | [119] |
| Vinasse | *Scenedesmus* sp. | - | 0.055 | 1.06 | 41.5 | - | Biomass | [82] |
| Slaughterhouse | *Chlorella vulgaris* | - | 0.575 | 1.15 | 85 | - | Biomass | [120] |

**Table 2.** *Cont.*

| Agro-Industrial Effluent | Microalgae | Medium Additives | Productivity (g L$^{-1}$ day$^{-1}$) | Biomass (g L$^{-1}$) | COD Reduction (%) | BOD Reduction (%) | Bio-Products | Reference |
|---|---|---|---|---|---|---|---|---|
| Slaughterhouse | *Chlorella vulgaris* | - | 0.433 | 1.3 | 19.4 | - | Chlorophyll-a (6.8 mg L$^{-1}$) | [121] |
| Slaughterhouse | *Chlorella vulgaris* | - | 0.194 | 5.4 | 94 | 99 | Biomass | [99] |
| Slaughterhouse | *Tetradesmus obliquus* | - | 0.234 | 6.6 | 96 | 99 | Biomass | [99] |
| Slaughterhouse | *Chlorella vulgaris* | - | 0.424 | 1.2 | 81 | - | Biomass | [97] |
| Slaughterhouse | *Chlorella sp.* and *Scenedesmus* sp. (1:1) | - | 0.033 | 0.231 | 13.4 | - | Biomass | [57] |
| Slaughterhouse | *Chlorella* sp. (Trebouxiophyceae) | 1% $CO_2$ at a flow rate of 0.5 L min$^{-1}$ | 0.030 | 0.18 | 49.4 | - | Biomass | [57] |
| POME | *Chlorella sorokiniana* and *Pseudomonas* sp. (1:1) | 70% (v/v) water; 200 mg L$^{-1}$ of glucose; 2.5% urea and glycerol 0.1 vvm $CO_2$ aeration. | 0.409 | 5.74 | 93.7 | - | Lipids(14.43% of the biomass) | [122] |
| POME | *Scenedesmus* sp. | - | 0.018 | 0.55 | 48 | - | Biomass | [86] |
| POME | *Chlorella pyrenoidosa* | 75% (v/v) waterurea and TSP (2:1) | - | - | 90.42 | - | Lipids(36% of the biomass) | [123] |
| POME | *Arthrospira platensis* | - | 0.032 | 0.19 | 15 | - | Biomass | [124] |
| POME | *Nannochloropsis* sp. | 20% Walne's medium | 0.05 | 0.7 | 47.5 | 47.2 | Lipids(45% of the biomass) | [125] |
| POME | *Chlorella* sp. | 50% BBM | - | - | 57.6 | - | Biomass | [126] |
| POME | *Chlorophyceae* (not identified) | 25% water | 0.047 | 0.7 | 89.6 | 99.4 | Biomass | [127] |
| POME | *Scenedesmus* sp. | - | 0.027 | 0.8 | 60 | - | Biomass | [86] |
| Cassava | *Desmodesmus subspicatus* LC172266 | 10% BBM 10% Trace elements solution | 0.041 | 0.63 | 89.04 | 85.85 | Lipids (21.40%) | [128] |
| Cassava | *Desmodesmus subspicatus* LC172266 | 10% BBM 10% Trace elements solution | 0.07 | 1.04 | 51.39 | 62.22 | Lipids (24.70%) | [128] |
| Cassava | *Desmodesmus armatus* | 10% BBM 10% Trace elements solution | 0.07 | 0.73 | 92 | 87 | Lipids | [94] |
| Cassava | *Chlorella sorokiniana* | - | 0.021 | - | - | 90 | Biomass | [95] |
| Cassava | *Scenedesmus* sp. | - | - | - | 72 | 74 | Lipids | [129] |
| Cassava | *Chlorella sorokiniana* P21 | - | 0.48 | 2.11 | 88 | - | Biomass (58% lipid) | [130] |
| Cassava | *Chlorella sorokiniana* P21 | - | 0.18 | 2.56 | 73.78 | - | Biomass | [130] |
| Cassava | *Chlorella sorokiniana* WB1DG | - | 0.049 | 1.3 | 63.42 | - | Biomass | [130] |
| Aquaculture | *Tetraselmis suecica* | 0.02g L$^{-1}$ N 0.01g L$^{-1}$ P | 0.068 | 0.9 | - | - | Biomass, carbohydrate (10.62%), lipids (25.06%) proteins (50.20%) | [63] |
| Aquaculture | *Chlorella vulgaris* | - | - | - | 71 | 55.72 | - | [113] |
| Aquaculture | *Chlorella vulgaris* | - | 0.465 | - | 98.10 | - | Biomass | Gao,2021 |
| Aquaculture | *Spirulina* sp. | 25% Zarrouk medium | 0.2 | 1.1 | 90 | - | Biomass, protein (63.73%), phycocyanin (16.60 mg ml$^{-1}$), polyunsaturated fatty acids (38.20%) and C18:3n6 (38.20%) | [131] |
| Aquaculture | *Chlorella sorokiniana* | - | 0.353 | 4.02 | 71.88 | - | Biomass, Lipids, Protein, Carbohydrate | [132] |
| Aquaculture | *Ankistrodesmus falcatus* | - | 0.16 | 2.25 | 61 | - | Biomass, Lipids, Protein, | [133] |
| Aquaculture | *Chlorella sorokiniana* | - | 0.107 | 1.51 | 69 | - | Biomass, Lipids, Proteins | [133] |
| Dairy | *Chlorella sorokiniana* | - | not show | 17 | 93 | - | Protein, lipids, biomass | [111] |
| Dairy | *Chlamydomonas polypyrenoideum* | 25% water | not show | 7.7 | 55.7 | - | Lipids (42%) | [134] |
| Dairy | *Chlorella* sp. | - | 0.000175 | 0.26 | 80.63 | - | Lipids | [109] |
| Dairy | *Chlorella sorokiniana* SU-1 | - | 1.96 | 0.153 | 67.6 | - | Lipids | [20] |
| Dairy | *Chlorella sorokiniana* | - | 1.667 | 0.108 | 57.17 | 56.95 | Biomass | [135] |
| Dairy | *Chlorella vulgaris* | - | 0.42 | 0.02 | - | - | Lipids | [136] |
| Dairy | *Chlorella* sp. T4 | 40% water | 0.0085 | 0.85 | 59.7 | - | Lipids | [137] |
| Dairy | *Chlorella vulgaris* | 25% water | 0.225 | 2.43 | 81.48 | - | Lipids | [138] |

## 4. Anaerobic Digestion as Pretreatment and Energy Recovery Strategy

### 4.1. Anaerobic Digestion as a Pretreatment Step

Anaerobic digestion (AD) is an effective waste treatment technology, especially for the wastewater or solid waste of high organic content. It can be used to recover organic compounds and produce biogas or other high-added-value products. Anaerobic systems are cheaper and relatively simple to operate but less efficient at removing organic matter than aerobic systems [54]. In contrast to conventional methods of landfilling or incineration, which are more polluting and do not allow the recovering of any material, anaerobic digestion has been used with the possibility of producing bioenergy and nutrient recycling [139]. The anaerobic digestion process decreases the organic waste load (both COD and BOD) [140]; organic matter is mainly converted into methane, while ammonia and phosphate are not consumed during the process [139]. The residual liquid effluent, called digestate, is the co-product generated in the AD process. It has a high content of nutrients (N and P) and cannot be released directly into nature [20]. Disposing and treating large amounts of digestate is an important problem to solve, as improper disposal can cause adverse environmental impacts. Due to the requirements of equipment and high energy consumption to reduce COD and other components of this digestate, nutrient recovery becomes attractive and has the potential for various applications [141,142]. Alternatively, some microalgae species can tolerate high nutrient content and other toxic pollutants from the digestate, and this can be used as a low-cost substrate for microalgae cultivation. Anaerobic digestion plays a vital role in the sustainable development of microalgae-based integrated biorefinery [143].

Wastewater systems significantly impact global water, energy, climate, and sustainability. Due to the high organic loads reported in effluents, such as POME, vinasse, cassava wastewater, dairy wastewater, slaughterhouses, and aquaculture, anaerobic digestion efficiently reduces this organic load. Still, it has the limitation of not removing phosphorus and nitrogen. Crude wastewater contains a very high concentration of organic compounds. It is characterized by high turbidity and low transparency, nutrients in the form of complex mixtures and is challenging regarding direct use by microalgae. The anaerobic digestion of wastewaters as a pre-treatment for the cultivation of microalgae has the advantage of the partial removal of organic compounds, besides the transformation of compounds into easily absorbed mineral forms and the reduction of turbidity, to improve the availability of light that enables photosynthesis [144].

The anaerobic digestion process mineralizes phosphorus and nitrogen into phosphate and ammonium, which are suitable forms for the assimilation by microalgae. The tertiary effluent generated after the cultivation of microalgae can be disposed of in an environmentally safe manner [145]. The integration of wastewater treatment plants and microalgae cultivation can be a resource for producing algae biomass, value-added compounds, and nutrient recycling [47]. This integration is also beneficial for the ability of microalgae to capture carbon dioxide and, consequently, reduce carbon dioxide emissions [146]. The $CO_2$ content present in biogas generally ranges from 20 to 60%, and microalgae tolerant to high $CO_2$ concentrations are preferred. Some microalgae may be more or less tolerant to the presence of $CH_4$, for example, a mutant *Chlorella* sp. MM2 tolerates up to 80% of $CH_4$, while a native strain *of Nannochloropsis gaditana* can tolerate 100% $CH_4$ without significant changes in biomass production and growth [147], meaning that biogas can be bubbled through culture media for $CO_2$ absorption (biogas upgrading is discussed in Section 5). The capacity of nutrient assimilation of these photosynthetic microorganisms reduces the risk of contamination of water bodies by effluent releases [148].

### 4.2. Secondary Effluents Composition

Raw effluents generally have physicochemical characteristics that limit their use directly in the production of microalgae, which is why pre-treatment steps of the effluents are necessary [72,149]. In the anaerobic digestion process, some compounds or nutrients are removed, mainly carbon sources, such as carbohydrates, and some lipids are practically

depleted. As a result, up to 90% of the DOC (dissolved organic carbon) is removed. [40,150]. The secondary effluent or digestate from the AD process is left with a residual COD and important nutrients, such as ammonia and phosphate. Volatile fatty acids, such as acetic, butyric, propionic, and lactic acids [151,152], are coproducts of AD and are also responsible for the residual COD of the effluent and can be used as alternative sources of carbon. In addition, complex polymers produced in AD can be bioavailable to be used or harnessed in other processes.

Dairy effluent presents high-polluting potential due to the high concentrations of organic matter and nutrients, such as potassium, nitrogen, and phosphorus, low pH, and high corrosiveness [153]. Dairy wastewater can be treated by integrating anaerobic digestion and microalgal remediation. Reports mention the COD removal of up to 80% in dairy effluent after anaerobic digestion [154]. Kusmayadi et al. [20] tested *Chlorella sorokiniana* SU-1 and the effect of inoculum sizes using dairy wastewaters. They found that AD effectively reduced the concentration of COD from 4000 to 978 mg $L^{-1}$ and total phosphorus content from 11.6 to 8.6 mg $L^{-1}$. Total nitrogen content was increased by anaerobic bacteria due to the transformation of organic nitrogen to $NH_4$. As microalgae primarily use inorganic nitrogen sources, this is particularly advantageous for subsequent microalgae cultivation. For vinasse and cheese whey, 89% COD removal efficiency can be obtained via the AD process [155]. The presence of nutrients in cassava wastewater, nitrogen and phosphorus, and carbohydrates characterize their potential as substrates in fermentative processes. They have a high COD of about 25,000 to 50,000 mg $L^{-1}$, and anaerobic digestion is commonly applied to decrease this load [156]. The effluent from slaughterhouses presents high organic loads (500–15,900 mg $L^{-1}$ COD) and a significant amount of nutrients, which are attractive to the treatment by anaerobic digestion. In studies with anaerobic digestion of slaughterhouse residues, removal percentages were obtained from between 16 and 22% [157] to 80% of COD [158].

Vadivelo et al. [57] reported microalgae cultivation in two raw, undiluted anaerobic digestates. A monoculture of *Chlorella* sp. and a mixed culture of *Chlorella* sp. and *Scenedesmus* sp. were found to be effective in treating these secondary effluents. The microalgae were cultivated in anaerobically digested municipal centrate and anaerobically digested abattoir effluent. *Chlorella* sp. Removed around 95% of $NH_3$-N and 58% of $PO_4^{3-}$, successfully demonstrating the use of the anaerobic digestates as culture media for microalgae.

The anaerobic digestion of aquaculture wastewater still concerns few studies, but its application has been considered an advantageous treatment alternative [159,160]. In cultivation systems that use recirculating aquaculture systems (RAS), the solid waste collected, mainly containing fecal matter and uneaten feed, can be anaerobically treated. Besides the reduction of solids and nutrients, there is a potential for methane production compared to other manure sources. Factors, such as the type of feed and species cultivated, impact the characteristics of the waste and its anaerobic biodegradability [161]. Diets rich in lipids can positively impact methane production, while lignocellulosic materials can limit the hydrolysis step of the anaerobic digestion process [161]. Table 3 presents certain secondary effluents commonly used for microalgae cultivation after AD.

**Table 3.** Physicochemical characteristics of secondary effluents from the anaerobic digestion of wastewaters that can be considered for microalgae cultivation.

| Parameter | SV | POME | CPW | AW | DW |
|---|---|---|---|---|---|
| pH | 4.3–7.9 | 7.64–7.8 | 7.5 | 6.9 | 6.8–7.5 |
| COD (g$O_2$ L$^{-1}$) | 4.9–33.16 | 2.57–6.4 | 0.235 | 0.303 ± 0.114 | 0.4–0.98 |
| Total suspended solids (g L$^{-1}$) | | 1.62–3.45 | 2.5 | | 0.1–0.13 |
| Total solid (g L$^{-1}$) | 3.4 | | - | | |
| Total nitrogen (g L$^{-1}$) | 0.49–3.56 | 0.25–0.34 | - | | 0.022–0.051 |
| Total phosphorus (mg L$^{-1}$) | - | 0.11–0.24 | - | | 8.6 |
| $NO_3^-$ (mg L$^{-1}$) | 298 | - | 0.6 | 140 | |
| $PO_4^{3-}$ (mg L$^{-1}$) | 34 | - | | 90 ± 8 | |
| $NH_4^+$ (mg L$^{-1}$) | | | 0.126 | 222 ±12 | |
| References | [162,163] | [149,164] | | [57] | [20] |

SV, sugarcane vinasse; POME, palm oil mill effluent; CPW, cassava-processing wastewater; AW, abattoir wastewater; and DW, dairy wastewater.

## 5. $CO_2$ Fixation and Biogas Upgrading

Beyond wastewater and solid residues in the agro-industry, discussions can expand to the issue of gaseous effluents and the development of bioprocesses that play an important role in the treatment, since they are considered one of the primary sources of emissions and contamination in the industrial process [81,165]. One of the emission sources originates from burning biomass or fossil fuels, where coal is widely used as an energy source [166–170]. Another source of emissions of gaseous effluents is biodigesters, which can be used to produce methane to generate electricity and heat and can emit high concentrations of $CO_2$ and $H_2S$ [171]. In this scenario of the biological treatment of gaseous effluent, microalgae represent a promising strategy for the photosynthetic fixation of $CO_2$ and other compounds (e.g., $H_2S$, sulfur oxides—SOx, and nitrogen oxides—NOx) present in the gas streams and offer the possibility of producing microalgae-based bioproducts and biofuels [165].

### 5.1. Microalgae Potential for Flue Gases Fixation

The photosynthetic apparatus of algae has been discussed as a potential strategy for GHG (greenhouse gas) mitigation and as a natural alternative for gaseous effluent treatment, since it simultaneously generates biomass that can be used in the recovery of high-value-added products. In this scenario, different microalgae species have been explored for industrial gas fixation to achieve greener processes and economic gain, strengthening the circular economy concept.

Most large industries rely heavily on fossil fuels, mainly coal and diesel, that emit a vast amount of flue gases, whose $CO_2$ concentration is typically above 10% [172–174]. Aside from $CO_2$ emissions, industrial flue gases are also often accompanied by other pollutants that aggravate the polluting potential of emissions, such as NOx, SOx, hydrocarbons, particulate matter, and trace pollutants. For instance, coal-burning generates flue gases with approximately 8 to 15% $CO_2$ content and varying concentrations of SOx and NOx, with reports of 30 to 200 ppm and 60 to 210 ppm, respectively [169,175–177].

The presence of these compounds in flue gas can be a challenge for microalgae-mediated gaseous effluent treatment because photosynthetic mechanisms (passive diffusion or enzymatic pathway [178,179]) responsible for $CO_2$ transport and assimilation in the cells are sensitive to changes in environmental conditions. Therefore, different microalgae species will have different biofixation potentials since they differ in their ability to tolerate the $CO_2$ concentrations and other compounds in the flue gas [180,181]. From another perspective, some genera can assimilate sulfur and nitrogen compounds in the flue gases as nutrient sources. There are reports of tolerance to SOx concentrations above 90 ppm by microalgae of the genus *Chlorella* that have been extensively studied for flue gas fixation [182].

In continuous mixotrophic cultivation conducted with municipal wastewater and coal-fired flue gas, *Chlorella vulgaris* showed a tolerance capacity of 180 ppm SOx and 250 ppm NOx, removing 59% and 55% of the compounds, respectively, and exhibited effluent treatment efficiency by reducing the organic matter concentration and $CO_2$ content (72%) [176].

Another aspect to be considered during flue gas biofixation is that the increase of $CO_2$ mole fractions in the inlet stream of the cultures leads to an intense reduction in pH, resulting from the dissolution of $CO_2$, NOx, and SOx in the aqueous medium, which induces the formation of acidic species [179,183]. This process causes metabolic inhibition, which can cause the cell death of the microalgae or can extend the lag phase [183]. To improve the practical application of gaseous effluent treatment methods, several species of microalgae with tolerance to high $CO_2$ concentrations have been investigated, in addition to efforts to elucidate the pathways for inhibition of the compounds present in the gases and strategies to minimize these problems [184–186].

It should be noted that tolerance and biofixation capacity are two different bioprocesses. Tolerating high flue gas concentrations will not necessarily result in efficient biofixation processes. The tolerance to different compounds and concentrations present in the flue gas was evaluated for microalgae species, with concentrations above 15% $CO_2$ [182,187,188]. However, higher biofixation capacities are reported at concentrations of approximately 10% $CO_2$ and have encouraged the development of strategies such as the intermittent injection of gas into the cultures or dilution of the gases, as reported by Yadav et al. [183], Moheimani et al. [169], and Matito-Martos et al. [189]. Another strategy used to minimize the toxic effect of high concentrations of $CO_2$ (15.6%), and in this specific case, the presence of hydrogen sulfide gas ($H_2S$) (120 ppm), is the use of scrubbers [187] and the serial connection of photobioreactors [187,190]. Recently, Chauhan et al. [180] demonstrated that the pre-acclimatization of microalgae could improve biofixation performance in the presence of SOx and NOx.

Developing mixotrophic cultivation strategies by integrating processes that use waste resources for microalgae growth and bioproduct recovery can reduce the operating costs of biological systems and improve process performance by increasing biomass productivity [191–194]. In addition, integrating $CO_2$ biofixation processes combined with wastewater treatment can provide more efficient treatment than individual operations [184,194]. This was observed in treating POME associated with $CO_2$ fixation using the microalgae *Chlorella* sp. and *Scenedesmus* sp. Furthermore, the authors suggested that the organic load present in the wastewater improved the stability of the pH even with $CO_2$ injection [184]. Table 4 presents recent studies that have conducted evaluations in microalgal cultures with wastewater and gas injection.

When a stable and well-established system is extant, it can be affirmed that mixotrophic cultures with wastewater and $CO_2$ injection favor the increase of biomass productivity due to the improvement of the photosynthetic process and the activation of metabolisms associated with the assimilation of organic carbon present in the wastewater [183]. The presence of other compounds or the restriction of nutrient sources can stress the cells and enhance the production of biomolecules such as carotenoids [180,188,192]. Song et al. [188] have shown that in the presence of 15% $CO_2$ and 80 ppm $SO_2$, the microalgae *Arthrospira* sp. increased pigment production compared to experiments without $SO_2$, demonstrating that the sulfur present in the flue gas acted as a nutrient source for the microalgae.

Developing flue gas biofixation integrated with wastewater treatment to obtain microalgal biomass is a complex process. Although the experimental analysis is especially relevant, technical and economic feasibility and environmental and social aspects are fundamental to evaluating the interaction of the different fields that comprise the process [193,195]. The recovery of bioproducts from microalgal biomass after wastewater treatment is an interesting strategy for developing an economically feasible approach that addresses the concept of a circular economy. There is a rapidly expanding market for products derived from microalgal biomass. However, when using wastewater and flue

gas in cultivation, the produced biomass may not be adequate for producing food, feed, or nutraceuticals. Still, it can be used for chemicals, biofuel, and biofertilizer production [193].

**Table 4.** Overview of reports that have evaluated integrated mixotrophic culture with wastewater treatment and $CO_2$ injection.

| Microalgae | Wastewater | Flue Gases | | | $CO_2$ Biofixation | Considerations | Reference |
|---|---|---|---|---|---|---|---|
| | | Origin | Composition | $CO_2$ Content Injected | | | |
| *Chlorella* sp. GD | Aquaculture | Natural gas | 8% $CO_2$ 57% NO | 8% $CO_2$ | 2333 mg $L^{-1}$ $d^{-1}$ | Growth is higher when aerated with flue gas compared to equivalent pure $CO_2$ | [179] |
| *Chlorella* sp. L166 | Soybean processing | Simulated | n.a. | 5% $CO_2$ | 25% | Diluted effluent (5x) and pure $CO_2$ injection | [196] |
| *Chlorella vulgaris* | Industrial wastewater (textile and food) | Coal-fired | 10% $CO_2$ 0.554% CO 61 ppm $NO_2$ 30 ppm SOx 9 ppm HC | 5% $CO_2$ | 187.65 mg $L^{-1}$ $d^{-1}$ | Increased lipid and carbohydrate accumulation | [183] |
| *Phormidium valderianum* BDU 20041 | Ossein effluent (Gelatin industry) | Coal-fired | 15% $CO_2$ | 15% $CO_2$ | 56.4 mg $L^{-1}$ $d^{-1}$ | High cell density to overcome metabolic stress | [197] |
| *Scenedesmus* sp. UKM9 *Chlorella* sp. UKM2 | Palm oil mill effluent (POME) | Simulated | n.a. | 10% $CO_2$ | 829 mg $L^{-1}$ $d^{-1}$ | Process conducted in two steps: effluent treatment and $CO_2$ fixation | [184] |

## 5.2. Biogas Upgrading

Biogas is a biofuel derived from the anaerobic digestion of biomass that has attracted attention in industries because it is a clean and economically low-cost energy generation process. The gas generated is composed chiefly of methane (40–70%) and $CO_2$ (20–60%), and other gases, such as nitrogen (0–3%), oxygen (0–1%), $H_2S$ (0–10,000 ppm), ammonia, and trace compounds [198–202]. The calorific value of biogas is limited to methane, which reduces the calorific value in the presence of other gases and leads to undesired emissions during combustion. Therefore, biogas upgrading is conducted to increase the calorific value of the gas by primary cleaning to remove $CO_2$ and $H_2S$ content, concentrating methane at the process output [203]. Any reduction in non-methane gases increases the calorific value of the biogas. However, biogas upgrading usually aims for a methane content higher than 85–90%.

Over the last decade, much has been explored concerning the microalgae-based upgrading process that could benefit biogas with high $CO_2$ content on a microalgae production and nutrient removal platform. Besides the $CO_2$ removal, other undesired compounds in the biogas can be removed—especially $H_2S$, which generates acid oxides during burning and leads to equipment corrosion [201,204].

Generally, biogas upgrading is conducted in absorption column reactors connected to photobioreactors for microalgal cultivation. Biogas is injected into the column, which contains a highly alkaline solution (pH > 9.0) and microalgae cells, where $CO_2$ and $H_2S$ are efficiently dissociated in the alkaline liquid phase of the column and a strongly pH-dependent reaction forms predominantly bicarbonate and hydrosulfides, respectively. [201,203]. The outflow of purified methane gas occurs from the top of the column, and the liquid is recirculated from the photobioreactor [205–210]. The oxygen presence can facilitate the oxidation of $H_2S$ to $SO_4^{2-}$, and these compounds can be a source of sulfur for the cells but can also inhibit growth when in concentrations above 200 ppm [201,211]. Frequently, removals higher than 95% of $H_2S$ are reported, which occur mainly via oxidation, resulting in systems with high absorption in the liquid solution ($SO_4^{2-}$) and secondarily through fixation by the microalgae cells, which play a crucial role in $CO_2$ fixation [200,201,207,212]. Table 5 presents $H_2S$ and $CO_2$ removal results in microalgae-based biogas upgrading processes.



**Table 5.** Biogas upgrading using microalgae-based biological systems.

| Biological Treatment | Biogas Content | | | | | CO$_2$ Removal (%) | H$_2$S Removal (%) | Reference |
|---|---|---|---|---|---|---|---|---|
| | Inflow (%) | | | Outflow (%) | | | | |
| | CH$_4$ | CO$_2$ | H$_2$S | CH$_4$ | | | | |
| Microalgae and bacteria consortium | 69.5 | 30 | 0.5 | 71 | | 88 | 100 | [200] |
| Microalgae and bacteria consortium | 69.2 | 32.7 | 0.12 | 87 | | 71 | 99 | [202] |
| Microalgae consortium (predominantly *Scenedesmus* sp.) | 68.7 | 21.6 | 0.012 | 50.4 | | 95 | 99.8 | [204] |
| Microalgae and bacteria consortium | 70.5 | 31.5 | 0.0005 | 97.3 | | 93 | 100 | [205] |
| Microalgae consortium (predominantly *Acutodesmus deserticola*) | 60 | 39.5 | 0.5 | 68.8 | | 20 | 100 | [206] |
| Microalgae consortium (predominantly *Pseudoanabaena* sp. and *Chlorella vulgaris*) | 60 | 38.7 | n.d. | 93.9 | | 90 | n.a. | [208] |

In microalgae cultivation, raceway photobioreactors are often used, which makes axenic cultivation difficult, and this encourages the use of microorganism consortia. Many reports evaluate the effect of symbiosis between bacteria and microalgae [202,208,213,214]. This technique has shown promising results at the laboratory scale and recently was assessed at a pilot scale with a consortium of microalgae and bacteria and showed efficiency higher than 90% for CO$_2$ and H$_2$S removal and methane content higher than 85%. [202,209]. Tubular reactors can be a strategy to improve biomass productivity in upgrading processes [210]. However, they are more expensive than open reactors, requiring greater control of parameters and engineering design complexity. Although axenic cultures are not frequently used, the microalgae of the genera *Scenedesmus* and *Chlorella* are often highlighted because of their versatility, tolerance to high CO$_2$ content (20–60%), and potential for capture. They have been reported to predominate even in non-axenic cultures. [207,208].

In addition to the above configuration, there is the possibility of using photobioreactors with biogas injection for CO$_2$ removal, where the culture is intermittently fed with biogas during the active photosynthesis phase [215–217]. To optimize biogas upgrading with direct injection into the photobioreactor, it is necessary to have sufficient residence time for CO$_2$ dissolution, uptake by the microalgae cells, and nutrient removal [203]. The optimization of multiple parameters can be fundamental to the performance of the reactor, such as luminosity that directly influences the growth rate of the microalgae and demonstrates impacts on nutrient removal when integrated with wastewater treatment [217].

Photosynthetic biogas upgrading in a commercial system must operate continuously with the control of CO$_2$ content, which ideally should be kept in a range of up to 6% and with low O$_2$ content to avoid a potentially explosive environment. There is a challenge in microalgae-mediated biogas upgrading, since higher CO$_2$ contents are present in biogas, which may require larger project areas [218].

Beyond that, oxygen is generated during photosynthesis, which can increase the content of biomethane [203]. One strategy applied to reduce the oxygen content of the biogas is recirculation during the dark phase of cultivation, where the photosynthetic step is interrupted and the increased respiration of the microalgae occurs [219]. In addition to environmental parameters, reducing the hydraulic retention time of biogas in reactors can provide engineering benefits and operational strategies with airlift configuration absorption unit with an evaluation of bubble dispersion and system behavior enabled short hydraulic retention time (10 min) that resulted in approximately 14% improvement in biogas calorific value with complete H$_2$S removal [206].

In the context of effluent treatment and biogas upgrading, the integration of biological processes has resulted in the co-culture of microalgae and fungi with emphasis on the fungus *Ganoderma lucidum*, which presents a high capacity for nutrient removal from effluents and friendly coexists in the environment with microalgae, having been recently evaluated for *Scenedesmus obliquus* and some species of *Chlorella* [220–222]. The technology

is still being assessed on a small scale and may result in higher energy expenditure and more unit operations. Even so, it shows promising results for nutrient (phosphorus, COD, nitrogen) and $CO_2$ removal at approximately 89% and 75%, respectively [222].

Recent reports have emerged aiming to improve the efficiencies of biogas upgrading processes for stable performance on a large scale. One of the strategies applied was to use oxidizing bacteria to act on the oxidation of $H_2S$ concomitant with $CO_2$ fixation by microalgae. This technique has shown promising results on a small scale and recently was evaluated at a pilot scale with a consortium of microalgae and bacteria and showed efficiency above 99% for removing $CO_2$ and $H_2S$ [202].

The impact of biogas upgrading integrated with wastewater treatment was evaluated by Rajendran et al. [218], who compared nine scenarios with different biogas upgrading technologies, which showed that microalgae-based upgrading technology would cost between 0.24 and 0.37 EUR/$m^3$. One challenge to the process scale-up is the requirement for reduced gas flow, compared to physical and chemical methods, in addition to the variation in yield and gas flow rate. This directly affects reactor design, area footprint, process control, and production cost [203,218]. It is worth emphasizing that the scale of the unit directly impacts the costs [223]. However, there is the possibility of cost reduction through the commercialization of bioproducts recovered from algal biomass in a circular economy approach that develops a process based on the management of waste with high environmental impact while expanding the commercial portfolio of industries for bioproducts from microalgae [224]. There are still gaps to be filled in terms of system optimization and scale-up that could reduce the costs of operation and production.

## 6. Tertiary Liquid Residues Destination and Water Reuse

Freshwater is a limiting resource in microalgae cultivation, and it is undeniable that the use of industrial wastewater is an interesting strategy to reduce the water footprint of these systems. It has already been demonstrated in the previous topics that besides improving the microalgae productivity and growth rate, there are concurrent processes of organic matter reduction present in the wastewater that can lead to serious environmental problems if disposed of inappropriately. Therefore, it is essential to consider that after cultivation, it is necessary to dispose of this wastewater safely, and environmental parameters must be considered.

However, there is a lack of research evaluating wastewater composition *after* microalgae cultivation. Most approaches are limited to assessing parameters required to comply with legislation for disposal into water bodies for the fertigation, or even recirculation, of the culture medium in algae cultivation, which will eventually also lead to disposal. The use of industrial wastewater as a culture medium for microalgae contributes to improving water quality for application in agricultural irrigation, often in compliance with environmental legislation in force and minimizing environmental impacts, or depending on fewer unit operations for downstream treatment, usually involving effluent polishing processes before discharge (e.g., pH adjustment; ultraviolet light; etc.) [225,226].

In a recent evaluation of wastewater composition after microalgae cultivation, Morais et al. [225] performed a comparative analysis of physicochemical parameters regarding current European laws, finding that the most significant impact of water reuse is inferred to pathogens, such as *Escherichia coli*. The limitations of reuse in urban areas were discussed, such as the necessary reduction of suspended solids and turbidity, as well as pH corrections, since the culture is conducted in an alkaline medium. In a non-conservative scenario, the water could be used for seed production (fertigation) [225]. The use for washing floors or animal stalls after the decantation and disinfection processes can also be a possible destination [216].

The recirculation of the medium in the algal culture represents a strategy for reducing the water input to the system and could be an insight into implementing "zero waste" processes [227,228]. Wastewater from aquaculture or shrimp farms can be strategically used in the microalgae growth to reduce the nutrient load and can subsequently be treated by filtration and disinfection to return to the animal cultivation process, reducing the water

footprint of the system, and acting in closed production cycles [228]. Industrial wastewater used in cultivation can also be used in recycling systems, as in a study by Daneshvar [227] with dairy effluent and the microalgae *Scenedesmus quadriculata* and *Tetraselmis suecica*. When recycling tertiary wastewater from cultivation, some losses in the quality of the microalgal biomass can occur, which affects the growth rate or changes in the biochemical compositions of the algae [227]. Despite being a potential strategy, when the primary wastewater is generated constantly and in large volumes by the industry, there may be a high treatment demand, making recycling unnecessary or even problematic.

Insights into value-added resource recovery are lacking in the field, and this step could play an important role in increasing microalgae-based processes' economic and environmental viability. One possible insight is the recovery of exopolysaccharides excreted by microalgae. These compounds possess bioactivity and can be applied in structuring an integrated system as building blocks for various commercial applications [229]. Exopolysaccharide recovery systems have already been evaluated in effluents such as POME and the microalgae *Phaeodactylum tricornutum* [230] and in systems with the microalgae *Arthrospira platensis* and *Chlamydomonas asymmetrica* with $CO_2$ injection in the culture [231]. The integration of processes with liquid effluents and gas injections to produce algal biomass and the recovery of exopolysaccharides can be targeted. The main challenge of these systems is the specificity of production from microalgae, which may limit the use of some species in integrated processes.

In an integrated project, industrial wastewater can be a source of carbon (combined with gas injection) in the mixotrophic cultivation of microalgae, which results in the recovery of bioproducts from biomass. In a zero-waste concept, the integration of a process to recover exopolysaccharides from the culture water could be proposed, provided that the engineering design is conditioned from the beginning to parameter adjustments for satisfactory yields. In addition, when all the possibilities are exhausted, water can be applied in washing systems, in fertigation, or destined for water bodies if following the adequate emission limits.

SDG 6 from the list of Sustainable Development Goals (SDGs) of the United Nations deals with increasing water use efficiency in all sectors by 2030 [232]. It emphasizes the relevance of studies that are concerned with evaluating the feasibility of reducing the use of freshwater in bioprocesses and with attention to the parameters of the wastewater generated from cultivation and, if possible, a feasible approach beyond microalgae cultivation and the recovery of algae-based bioproducts.

## 7. Fate of Xenobiotics and Heavy Metals

The use of microalgae in the bioremediation of xenobiotic compounds is a promising technology due to the microalgae's capacity to eliminate different types of compounds, including emerging ones [233]. Besides nutrients, such as carbon, nitrogen, phosphorus, and potassium, which are considered contaminants due to their potential impact on water bodies, other pollutants have been removed using microalgae: heavy metals, pharmaceutical compounds, and radioactive compounds, among others [234].

Microalgae cultures remove xenobiotic compounds using four main mechanisms: (i) biodegradation (Figure 2A), in which decomposition reactions reduce the toxicity of the pollutant; hydrolysis and oxidation reactions generally predominate, and the process can be intra- or extracellular [235]. This mechanism was observed in the microalgae *Scenedesmus obliquus* and *Chlorella pyrenoidosa* in the biodegradation of pharmaceutical compounds [236]. A second mechanism is (ii) bioaccumulation, when the pollutant accumulates inside the cells, which can change their structure (Figure 2B). The microalgae *Desmodesmus subspicatus* employs this mechanism in the bioremediation of radiolabeled 17a-ethinylestradiol. Researchers [237] also observed the bioaccumulation mechanisms in the carbamazepine bioremediation process using the microalgae *Chlamydomonas mexicana* and *Scenedesmus obliquus*. Additionally, some metals, such as Cu, Pb, Cd, and Co, can be eliminated by *Cladophora glomerata* and *Oedogonium rivulare* by short-term uptake and others, such as

Ni, Cr, Fe, and Mn, in continuous uptake. The third mechanism is (iii) bioadsorption, where the pollutant attaches to the cell surface through electrostatic forces (Figure 2C). According to [235], microalgae cell walls are negatively charged due to the predominance of functional groups, such as carboxyl and phosphoryl, compared to amines. They also contain different polymers, which also facilitate the adsorption process. This mechanism was observed in dead *Scenedesmus obliquus* and *Chlorella pyrenoidosa* biomasses in removing progesterone and norgestrel [236]. Because bioadsorption is extracellular, the process is affected by the system's hydrophobicity and the pollutants' functional groups. Thus, xenobiotic compounds with cationic groups are highly attracted to the microalgae surface through electrostatic interactions [235]. The fourth mechanism for bioremediation is iv) bioaugmentation, based on the use of two or more species of microalgae or other types of microorganisms (Figure 2D). The main advantage of this bioremediation mechanism is that it generates synergy between the different microorganisms in the pollutant degradation process, in addition to expanding the spectrum of pollutants to be bioremediated. For instance, the symbiosis between *Chlorella protothecoides* and *Brevundimonas diminuta* promotes nutrient removal from wastewater. One of the approaches used for the bioaugmentation process is based on $CO_2$ and $O_2$ gas exchange in the effluent. However, the population interactions are not fully understood [238].

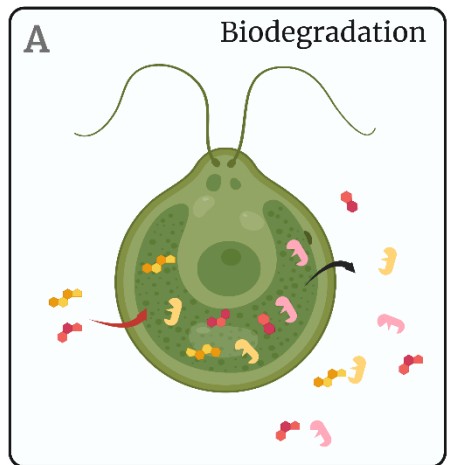
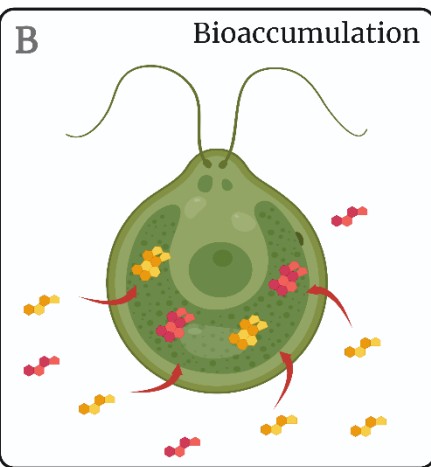
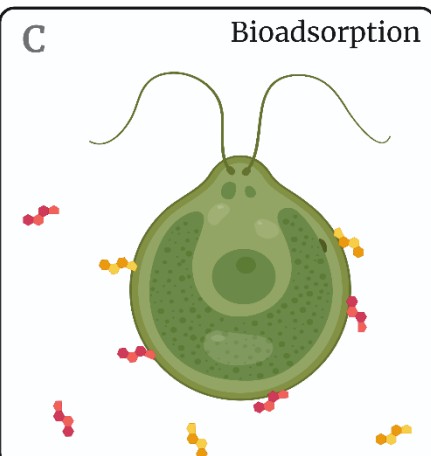
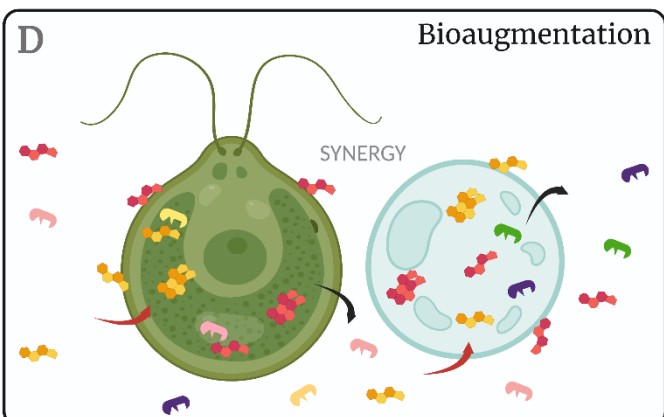

**Figure 2.** Bioremediation mechanisms at play in microalgae cultures. (**A**) Biodegradation, where the cells directly degrade xenobiotics into innocuous molecules; (**B**) bioaccumulation, where the xenobiotics are not degraded but accumulate in living cells and can be removed with them; (**C**) bioadsorption, where molecules cling to the surface of living or dead cells and can be removed with the biomass; and (**D**) bioaugmentation, where pollutant molecules are degraded in an exchange of metabolites between microalgae and one or more symbionts.

## 8. Microalgae Products

Microalgae can be employed in the production of different commercially important biomolecules. Their potential contribution to the global bioeconomy includes a wide range of bioproducts, such as biofuels (bioethanol, biohydrogen, and biodiesel), bioplastics, biofertilizers, lipids, carotenoids, nutraceutical biomass, animal and human nutrition products, and others [239]. Bioremediation, nutrient recycling through wastewater treatment, and carbon fixation are generally carried out during the production process, which is an excellent advantage of these technologies [240]. Other benefits are the high growth speed, low production cost, high adaptability to a wide range of growth conditions, pH, and climate, a faster cycle of cultivation, processing, and harvesting, and the ability to grow in wastewater [241]. Another advantage is that microalgae can be produced using effluents and wastewater in high volumes and areas unsuitable for agriculture. A high diversity of photobioreactors models has been developed to produce different bioproducts, which creates new environmentally friendly solutions for the industry. However, some challenges include low biomass production, acclimatization, contamination (depending on the bioreactor model), harvesting, and downstream processes' costs. Some recent works applied the biorefinery concept to convert the different fractions of algal biomass into high-value-added products [242]. In this context, microalgae biomass is integrally exploited as a substrate in the so-called zero waste-technology approaches where pretreatment and microbial fermentation techniques can be employed.

Bioplastics are alternative plastics obtained from renewable sources [243]. They can be produced from microalgae through different methods: (i) the blending of synthetic polymers and bioplastics with microalgae, such as proteins, starch, cellulose, and PHA [244,245]; (ii) biopolymers produced from microalgae cells [246] involving the intracellular accumulation of biomolecules, such as PHA, PHB, and starch, under certain conditions of nutrient deficiency, and its subsequent extraction [242]; and (iii) the conversion of microalgae into construction blocks suitable for polymerization [247]. *Chlorella* sp. and *Spirulina* sp. are the main microalgae species employed for bioplastic production. Genetic modifications in microalgae species have also been studied to improve the production of PHAs [248].

Microalgal lipids and fatty acids are important bioproducts that have attracted interest due to their high content of unsaturated lipid molecules. They can be used as feedstocks for biofuels and biomedical applications [249]. The content of lipids in oleogenic microalgae can vary from 30–70% of the weight of dry cell biomass [250]. Omega fatty acids ($\omega-3$ and $\omega-6$) are accumulated within microalgae cells through different environmental and/or nutritional stress [239]. However, only a small percentage of microalgae species can accumulate omega fatty acids. Milano et al. (2016) investigated biodiesel production from microalgae strains with high concentrations of oleic acid in their fatty acids. The lipid content of algal biomass affects the quality and presentation of biodiesel and its suitability as an alternative fuel to petroleum-based diesel [251]. Genetic engineering techniques and molecular biotechnology are applied to increase lipid productivity in microalgae for oil conversion and use for biofuel production.

Microalgae are a natural source of carotenoids, with β-carotene (provitamin A), astaxanthin, and lutein being the main commercially valuable carotenoids found in microalgae [252]. *Chlorella vulgaris*, *Chlorella protothecoides*, *Scenedesmus almeriensis*, *Dunaliella salina*, *Haematococcus pluvialis*, *Porphyridium cruentum*, and Haslea ostrearia have been reported for carotenoid production. Because it can be produced throughout the year, carotenoid degradation by long-term storage is eliminated [253].

The application of microalgae in biofuel production occurs through the photosynthesis process performed by most microalgae, which allows $CO_2$ and solar energy to be converted into sugar and biomass [241]. Microalgae are the most potent species in biofuel generation and wastewater treatment compared to plant-based biofuel crops because they use them as sources of nitrogen, phosphorus, and micronutrients for growth [254,255]. Microalgae have high levels of different carbohydrates, such as glycogen, starch, agar, and cellulose, which can be converted into fermentable sugars for bioethanol production. Some carbohydrate-

rich microalgae, such *as Chlamydomonas reinhardtii* and *Chlorella vulgaris,* have the potential to produce carbohydrates for bioethanol production [256]. In biohydrogen production, microalgae have been considered an inexpensive potential source. It was reported that the hydrogenase enzyme is responsible for biohydrogen production, in sulfur-deprived, selected microalgae species. In the generation of biogas from microalgae, the challenge is related to the raw material, seasonal variation, and the pretreatment of algal biomass [257]. The production of biogas from microalgae through anaerobic digestion has the advantages of using wastewater, the feasibility of nutrient recycling, maximum biomass utilization, lower operating costs, and lower energy consumption [258].

Wastewaters from various processing streams can produce algal biomass for different applications and bioproducts (Table 6).

**Table 6.** Microalgal products from different residues.

| Product | Type of Effluent | Microorganism | Production/Yield | Reference |
|---|---|---|---|---|
| Allophycocyanin | Swine wastewater | *Thermosynechococcus* sp. CL-1 | $12,07 \pm 0,3\%$ dwc | [259] |
| Algal biomass | Dairy effluent | *Chlorella vulgaris* | $0.175$ mg $L^{-1}$ day$^{-1}$ | [260] |
| Biomass for bioethanol production | Domestic wastewater | *Scenedesmus sp.* | Maximum biomass productivity: $62$ mg $L^{-1}$ day$^{-1}$ | [261] |
| Biomass for bioethanol production | Domestic wastewaters | *Scenedesmus sp.* | Biomass $0.84$ g $L^{-1}$; lipid productivity $8.6$ mg $L^{-1}$ day$^{-1}$ | [262] |
| Biomass for bioethanol production | Dairy wastewater | *Scenedesmus obliquus and Chlorella vulgaris* | Dry biomass produced was in the range of $2.30$ to $3.10$ g $L^{-1}$; yields for lipids $0.068$ g $L^{-1}$ day$^{-1}$ and carbohydrates $0.114$ g $L^{-1}$ day$^{-1}$ | [263] |
| Biomass for biodiesel production | Cheese whey wastewater | *Chlorella pyrenoidosa* | Maximum algal biomass yield: $2.44$ g $L^{-1}$; Lipid productivity: $77.41$ mg $L^{-1}$ day$^{-1}$, | [143] |
| Carotenoids | Aquaculture effluent | *Spirulina* sp. LEB 18 | $9.68$ μg mL$^{-1}$ | [264] |
| Lipid | Aquaculture wastewater | Microalgal consortium of *Euglena gracilis* and *Selena strum* | $84.9$ mg $L^{-1}$ | [260] |
| Lipid | Dairy effluent | *Arthrospira platensis* | $158$ mg $L^{-1}$ day$^{-1}$ | [260] |
| Lipid | Dairy effluent | *Scenedesmus* sp. ASK22 | $30.7\%$ content | [260] |
| Lipid | Molasses wastewater | | $92.33\%$ content | [260] |
| Lipids | POME wastewater | *Chlorella* sp. | $66\%$ | [265] |
| PHB | Sewage wastewater | *Botryococcus braunii* | $247$ mg $L^{-1}$ of PHB | [266] |

## 9. Circular Bioeconomy in Microalgal Production from Agro-Industrial Wastes

Microalgae represent a promising strategy in the circular bioeconomy because of their ability to fix $CO_2$ and to be cultivated in non-arable lands using aqueous residues, reducing their polluting potential, and recovering nutrients in their biomass. Besides promoting significant biochemical transformations that result in environmental benefits, microalgae are an important source of various bioproducts, including edible oils, biofuels, biopigments, nutritional protein, peptides, and biofertilizer, which in turn can enhance the economic performance of microalgal biorefineries.

The market value of some of the most important autotrophic microalgal species—*Spirulina*, *Chlorella*, *Dunaliella,* and *Haematococcus*—is between EUR 370 and 500 million, corresponding to an annual production of 11.180 tons/year to 15.950 tons/year and an average price of 227–530 EUR/kg on dry basis [267].

The diffusion of microalgal technologies in circular processes can promote significant advances in establishing a bio-based economy. Important indicators to be considered in this context include the life-cycle assessment (LCA) evaluating environmental impacts, such as global warming potential (GWP), fossil energy requirement (FER), greenhouse gas (GHG) emissions, water footprint, land transformation and use, abiotic depletion (ADP), eutrophication potential (EP), acidification potential (ACP), ozone depletion (OD), photochemical oxidation potential (POP), and toxicity parameters. Banu et al. [268] summarized the life-cycle analysis and environmental impacts of different microalgal biorefineries configurations. The most frequently analyzed environmental impacts were GHG (mainly $CO_2$) emissions and GWP associated with microalgal biomass and biodiesel production. For example, the "well to tank" LCA reported by Monari et al. [269] to produce 1 MJ of biodiesel using *Nannochloropsis* sp. cultivated in photobioreactors showed that the use of wastewater is determinant to avoiding GHG emissions and energy demand. Otherwise, the microalgal biodiesel produced by conventional technology (i.e., using freshwater) had higher energy demand and GHG emissions than fossil diesel.

The evaluation of microalgal biorefineries from the economic point of view can be carried out through the analysis of life-cycle costing (LCC). It represents the aggregation of energy, installation, downstream, operating, maintenance, environmental, and decommissioning costs over the complete lifetime of the product and considers the operating cost (OC), the salvage value (SV), the maintenance cost (MC), and the capital cost (CC) [268]. Using agro-industrial wastewater in microalgal cultures reduces the operating costs associated with raw materials (water and nutrients). However, the presence of insoluble solids and undesired substances may require additional processing steps that can impact the economic performance of the process. It is challenging to elucidate the economic impact of agro-industrial wastewater in microalgal biorefineries since most studies focus on major process steps and equipment, such as downstream unit operations, cultivation modes, and bioreactor types. For example, the economic performance of microalgal biodiesel production (9.3 million gallons/year) using autotrophic mixed cultures was highly dependent on the reactor type. When comparing open ponds with closed tubular photobioreactors, the cost of lipid production to achieve a 10% return was USD 8.52/gal versus USD 18.10/gal respectively; when considering a hydrotreatment for fuel upgrade, the values rose to USD 9.84/gal and USD 20.53/gal, respectively [270]. Similarly, the operational costs of β-carotene and fertilizer production using *Dunaliella salina* increased from ca. 7.9 million EUR/year in open paddlewheel ponds to 13.2 million EUR/year in closed photobioreactors [271]. Economic studies focusing on agro-industrial wastewaters versus defined media should be developed, at least at a pilot scale, to demonstrate the viability and scalability of such processes.

The production of microalgal biomass and bioproducts in agro-industrial wastewaters still needs further development to reduce production costs so that these bioproducts can become economically competitive. The long cultivation time and the low yields are among the most cited obstacles. In a review of the potential of using swine, cattle, and poultry wastewater in microalgal cultures, Lopez-Sánchez et al. [272] pointed out that, although promising, the production of biomass using wastewater in a circular economy perspective is still limited to low yields because it depends, among other things, on the adaptability of the strain to the physicochemical conditions of the medium, which often needs to be sterilized.

Although there are challenges concerning process economics, the technical viability of microalgae cultivation using circular bioeconomy approaches has been demonstrated at laboratory-, pilot-, and large-scale. Lopez-Pacheco et al. [273] presented studies on the use of wastewater from swine farms in the cultivation of different species of microalgae both as wastewater treatment and to produce byproducts of commercial interest. Ciardi et al. [274] cultivated *Scenedesmus almeriensis* using pig slurry at 5%, reaching a biomass productivity of $0.68$ g $L^{-1}$ $d^{-1}$. García-Galan et al. [275] studied the treatment of agricultural runoff from irrigation channels in rural areas in a full-scale horizontal tubular system with *Pediastrum* sp.,



*Chlorella* sp., *Scenedesmus* sp., and *Gloeothece* sp. The authors observed a 22.8% increase in biomass production in 4 days, with up to 72% removal of various organic pollutants.

Examples of pilot- to large-scale facilities where microalgae are cultivated in wastewaters applying circular economy approaches were summarized by Goswami et al. [276]. MaB-floc SBRs (microalgae-bacterial flocs in sequencing batch reactors) in Belgium used aquaculture wastewater (ca. 12 m$^3$, 40 mg L$^{-1}$ N, 2 mg L$^{-1}$ P) to cultivate microalgae and bacteria in outdoor raceway reactors, with a biomass yield of 25 g m$^{-2}$ day$^{-1}$ and a nutrient removal efficiency above 50% for nitrogen and phosphorus. Researchers at Chilgokgun Agricultural Technology Center in South Korea cultivated *Chlorella* sp., *Coelastrella* sp., *Acutodesmus* sp., and *Pseudopediastrum* sp. in outdoor raceway ponds with different wastewaters (ca. 236,000 m$^3$, 15–30 mg L$^{-1}$ N, 3–6 mg L$^{-1}$ P), producing 2612.7 kg biomass with nutrient removal efficiencies of 13.1 mg/kg biomass for nitrogen and 2.5 mg/kg biomass for phosphorus. NPDEAS, at the Federal University of Paraná in Curitiba, Brazil, treated 9.38 m$^3$ of swine slaughterhouse wastewater in a closed photobioreactor producing 1.5 kg m$^{-3}$ day$^{-1}$ of microalgal biomass. The development of a microalgal biorefinery to be integrated into an ethanol industry was reported by Sydney et al. [81]. Sugarcane vinasse was used as a medium for chemical and biological $CO_2$ fixation, the first achieved by the reaction of the gas with alkalinized vinasse to form carbonate and bicarbonate salts and the second by the cultivation of different autotrophic microalgal and cyanobacterial species, namely *Botryococcus braunii*, *Synechococcus nidulans*, *Chlorella kessleri*, *Chlorella vulgaris*, *Neochloris oleoabundans*, *Scenedesmus obliquus*, *Arthrospira platensis*, *Arthrospira laxissima*, and *Arthrospira maxima*. The previous chemical treatment was important to reduce turbidity and organic load, allowing the cultivation of the microorganisms at vinasse concentrations between 70 and 100%, with a biomass production of 2.25 g L$^{-1}$ of *B. braunii* reached after 15 days.

## 10. Process Development and Perspectives

Coupling microalgae production with residue treatment is conceptually simple but operationally complex, because of the variable composition of wastewaters and the process parameters involved in massive microalgae cultures. However, there are vital aspects that, if carefully considered, can aid in process development. The key elements to be considered are effluent composition, availability, and the microalgae products of interest. From these initial constraints, the options can be narrowed down to a few process designs that can be compared on a techno-economical basis. Such considerations are complex and case-specific, but a few initial concerns are:

Production system: Residue volume and biomass use dictate the type of bioreactor to be used. Relatively small volumes can be processed in closed photobioreactors in a more controlled process that can lead to specialty algal products, since the risk of contamination is low and the quality of the product can be guaranteed. Large volumes, however, require large processing areas. In this case, open bioreactors are more cost-effective; however, the culture can have contaminants. This reduces the possible applications, unless downstream and fractionation guarantees the removal or inactivation of biological contaminants.

Residue COD and composition: Recovering energy as biogas can be profitable for the industry and will also lead to a digestate more adequate for autotrophic microalgal growth. However, dilute residues can be directly transferred to microalgal production. When a high amount of suspended or macromolecular solids is extant, as with CPW, minimal treatment (flocculation, removal of suspended solids) can recover the organic load for biodigestion while giving a clarified, less turbid supernatant effluent. The BMP, biomethane potential of residues, can be estimated at around 350 NL$_{CH4}$. kg $_{COD}$$^{-1}$, assuming a substrate such as a carbohydrate or lactic or acetic acid with minimal formula $CH_2O$ is present; this initial estimation can be used to define whether it is worth including an AD step in the process.

Most secondary and tertiary residues also have large amounts of N- and P, but in imbalanced quantities, leaving unconverted nutrients. If the objective is simply to take advantage of excess nutrients, then the medium can be left unchanged. However, to

maximize biomass production, a first improvement is adding a nitrogen or phosphate source to approach the Redfield ratio of 16:1 N to P (molar basis). The medium composition can be further adjusted for specific microalgae.

Residue toxicity and pathogenicity is a final concern that must be evaluated. The toxicity here refers to microalgae and cyanobacteria, which can be affected by cyanide from CPW, high salinity, concentrated transition metals such as $Cu^{2+}$, and antibiotics or pesticides. Partial pretreatment and microalgae selection can tackle these problems. Still, these also affect the final biomass use—it is recommended that an orthogonal approach is used, i.e., the product from one agro-industry chain (e.g., vinasse) is applied in another chain (e.g., animal rearing) or is not cycled back to the field (e.g., microalgal biomass for energy can be simply co-fired). Fractions such as biodiesel, carotenoids, or bioactive extracts can guarantee the removal of residual xenobiotics and pathogens through downstream processing steps.

Microalgae selection: This critical step affects both productivity and process resilience. The options for specialty products, such as nutritional biomass, are few and depend on the envisaged market. Still, among a specific genus (e.g., *Arthrospira*), a few species and many strains can be tested to find a robust culture to grow in the residue. Adapting the culture is essential: inoculating a reactor cannot be simply carried out with a pure culture, which would lead to a shock. Slowly rising the effluent concentrations allow the culture to adapt, but even so, there may be a limit, thus requiring residue dilution.

When there is no product constraint, and the objective is to recover nutrients or reduce the effluent COD and eutrophication potential, a culture can be selected for its suitability to grow in the residue with high productivity. Such a selection is based on sampling microalgae from a similar environment and using classical microbiological techniques, careful work that pays by taking advantage of the enormous natural biodiversity.

Microalgae must be selected for their rapid growth and resistance to contamination, especially in open photobioreactors. An approach that is becoming more popular is using polycultures and microalgal-bacterial consortia, at least for biomass, that will be recycled for bioenergy production. This takes advantage of the adaptability of a mixed culture throughout the year and is especially suitable for simple open systems.

Stepwise development: Coupling microalgae production with agro-industry waste recycling is not complicated, but it is complex—in the sense that many factors are interlinked in the process, from residue and weather variability to the management and downstream processing of massive microalgal cultures. Therefore, a sensible approach is stepwise process development, starting with the laboratory evaluation of the suitability of residues to AD and algae growth, using microalgae from culture collections and even isolates or mixed cultures. The process can be conceptually evaluated from laboratory data through classical techno-economic analysis and, if seemingly promising, it can be developed as a pilot scale. From this proof of concept, it can be safer to develop an industrial process. Microalgae production can also be developed stepwise: even if the final aim is to produce a specialty microalgal fraction, starting with a robust or fast-growing culture such as *Arthrospira* or *Chlorella*, simply separating and drying biomass or recycling it to AD can allow the workforce to focus on production issues, and in time, other microalgae and more sophisticated fractionation processes can be tested.

**Author Contributions:** Conceptualization, J.C.C. and C.R.S.; writing—original draft preparation, J.C.C., D.T.M.-A., W.J.M.-B., S.G.K., M.C.M., A.B.P.M., C.R., T.S., L.P.S.V., S.V., A.L.W. and V.T.S.; writing—review and editing, J.C.C., S.G.K., A.B.P.M., C.R., L.P.S.V., A.L.W., V.T.S. and C.R.S.; visualization, T.S. and W.J.M.-B. All authors have read and agreed to the published version of the manuscript.

**Funding:** The researchers were funded by the Coordination for the Improvement of Higher Education Personnel, CAPES foundation—PROEX Program, and the National Council for Scientific and Technological Development, CNPq, grant no.313276/2021-8.

**Institutional Review Board Statement:** Not applicable.

**Informed Consent Statement:** Not applicable.

**Data Availability Statement:** Not applicable.

**Conflicts of Interest:** The authors declare no conflict of interest.

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
