# Peer review of "Agro-Industrial Wastewaters for Algal Biomass Production, Bio-Based Products, and Biofuels in a Circular Bioeconomy"

_fermentation, doi:10.3390/fermentation8120728_

Round 1
Reviewer 1 Report
The review "Agro-industrial wastewaters for algal biomass production, bio-based products, and biofuels in a circular bioeconomy" is referred to an interesting and up-to-date subject. In general, the paper is well-written and well-organized. In my opinion, authors should stress what more this review offers compared to other reviews made on this subject.
Some comments
1. Abstract: Authors should mention the specific wastewaters that are mentioned and discussed in this review.
2. Keywords: Please avoid abbreviations.
3. Lines 92-95: Authors should also mention the intense color of wastewaters.
4. Figure 2: Please illustrate more clearly Figure 2 and give more details in its caption.
5. Abbreviations throughout the text are not defined. For example, FDA's GRAS (line 37), CAGR (line 40), CAPEX (line 61), PC and AX (line 90), UASB (line 221), HVAC (line 826) etc.
6. Please correct subscripts and superscripts in the chemicals throughout the text. For example, KH2PO4 (line 271) should be changed to KH2PO4 etc.
Author Response
We thank the reviewer for his or her time and effort spent in evaluating our manuscript. We addressed all suggestions (which are copied below) and comment actions taken (after each suggestion)
Reviewer #1
The review "Agro-industrial wastewaters for algal biomass production, bio-based products, and biofuels in a circular bioeconomy" is referred to an interesting and up-to-date subject. In general, the paper is well-written and well-organized. In my opinion, authors should stress what more this review offers compared to other reviews made on this subject
Answer: thanks for a generally positive review. This is a broad review of the recent literature, focused on references of the last five years. However, we imagine that any reader will expect that a review brings up-to date information, and opted by not stating this in the abstract. We stressesd, as suggested, the main residues discussed, and the point of discussing circularity and process integration.
Some comments
Abstract: Authors should mention the specific wastewaters that are mentioned and discussed in this review.
Answer: We now included this information in the abstract.
Keywords: Please avoid abbreviations.
Answer: We agree that abbreviations should be avoided, but POME, similarly to BOD or MSW, are very common acronyms in their respective niches and could aid in the paper indexation.
Lines 92-95: Authors should also mention the intense color of wastewaters.
Answer: Good point. We included this aspect where suggested.
Figure 2: Please illustrate more clearly Figure 2 and give more details in its caption.
Answer: We revised figure 2 to improve communication and detailed its caption.
Abbreviations throughout the text are not defined. For example, FDA's GRAS (line 37), CAGR (line 40), CAPEX (line 61), PC and AX (line 90), UASB (line 221), HVAC (line 826) etc.
Answer: We manually searched for acronyms and abbreviations that were not defined in the first instance. We described these on the first occurrence and repeated the definitions in some critical text parts.
- Please correct subscripts and superscripts in the chemicals throughout the text. For example, KH2PO4 (line 271) should be changed to KH2PO4etc.
Answer: We searched for all instances with a character or parenthesis and a number to correct superscripts and subscripts. We did not correct sub- or superscripts in references because these were generated using a reference manager. Indeed, there were about 17 inconsistencies in the text, some due to formatting.
Reviewer 2 Report
1. A specific term with abbreviation shown in the first time should be completely spelled out. For example, I guessed “CAPEX” in L. 61 might be “capital expenditure”. Same as PC and AX in L. 90.
2. L. 77. “The liquid fractions are the primary effluents”. It should be clearly describe effluents from what?
3. What is “secondary residues” seen in L. 79?
4. L. 81 – 82. There were several published articles indicating algal growth with flue gases. The following review article could be referred to:
Huang et al., 2016. Current Techniques of Growing Algae Using Flue Gas from Exhaust Gas Industry: a Review. Applied Biochemistry and Biotechnology 178: 1220-1238.
5. L. 83. I would suggest the sentence be rephrased as “can be the biogas produced in anaerobic digestion”.
6. The period after “biofuel” in L. 91 should be deleted.
7. L. 211. What did “difficult sunlight or artificial light” mean? I guess that the authors should mean the high color and turbity of the vinasse hindered the irradiation of sunlight or artificial light into the culture medium.
8. The authours should check the spelling and writing throughout the manuscript. A language editing service is recommended.
Author Response
We thank the reviewer for his or her time and effort spent in evaluating our manuscript. We addressed all suggestions (which are copied below) and comment actions taken (after each suggestion)
Reviewer #2
- A specific term with abbreviation shown in the first time should be completely spelled out. For example, I guessed "CAPEX" in L. 61 might be "capital expenditure". Same as PC and AX in L. 90.
Answer: We manually searched for acronyms and abbreviations that were not defined in the first instance. We described these on the first occurrence and repeated the definitions in some critical text parts.
- L. 77. "The liquid fractions are the primary effluents". It should be clearly describe effluents from what?
Answer: We revised the text to clarify the terms "primary" and "secondary" effluents. We also added vinasse as an example and changed the fragmented sentence in L. 79.
- What is "secondary residues" seen in L. 79?
Answer: We corrected to secondary effluents and revised the whole paragraph to improve communication.
- L. 81 – 82. There were several published articles indicating algal growth with flue gases. The following review article could be referred to:
Huang et al., 2016. Current Techniques of Growing Algae Using Flue Gas from Exhaust Gas Industry: a Review. Applied Biochemistry and Biotechnology 178: 1220-1238.
Answer: Thank you for the suggestion; it is a good short review.
- L. 83. I would suggest the sentence be rephrased as "can be the biogas produced in anaerobic digestion".
Answer: Corrected as suggested.
- The period after "biofuel" in L. 91 should be deleted.
Answer: The whole sentence was revised and corrected.
- L. 211. What did "difficult sunlight or artificial light" mean? I guess that the authors should mean the high color and turbity of the vinasse hindered the irradiation of sunlight or artificial light into the culture medium.
Answer: Indeed, that is what we meant. We now simplified the sentence.
- The authours should check the spelling and writing throughout the manuscript. A language editing service is recommended.
Answer: The manuscript was thoroughly revised.
Round 2
Reviewer 2 Report
The manuscript has been revised very well. I have no further comment.